# Effects of building resilience skills among undergraduate medical students in a multi-cultural, multi-ethnic setting in the United Arab Emirates: A convergent mixed methods study

**Farah Otaki[1,2]‡, Samuel B. Ho[3,4]‡\*, Bhavana Nair[5], Reem AlGurg[1,3], Adrian Stanley[3,4], Amar Hassan Khamis[6], Agnes Paulus[2,7], Laila Alsuwaidi[3,5]**

**1** Strategy and Institutional Excellence, Mohammed Bin Rashid University of Medicine and Health Sciences, Dubai Health, Dubai, United Arab Emirates, **2** Department of Health Services Research, Faculty of Health, Medicine, and Life Sciences (FHML), Care and Public Health Research Institute (CAPHRI), Maastricht University, Maastricht, The Netherlands, **3** College of Medicine, Mohammed Bin Rashid University of Medicine and Health Sciences, Dubai Health, Dubai, United Arab Emirates, **4** Mediclinic Middle East, Dubai, United Arab Emirates, **5** Student Affairs, Mohammed Bin Rashid University of Medicine and Health Sciences, Dubai Health, Dubai, United Arab Emirates, **6** Hamdan Bin Mohammed College of Dental Medicine, Mohammed Bin Rashid University of Medicine and Health Sciences, Dubai Health, Dubai, United Arab Emirates, **7** Faculty of Health, School of Health Professions Education (SHE), Medicine, and Life Sciences (FHML), Maastricht University, Maastricht, The Netherlands

☙ These authors contributed equally to this work.
‡ These authors share first authorship on this work.
\* samuel.ho@mbru.ac.ae

## Abstract

### Introduction

Although curricula teaching skills related to resilience are widely adopted, little is known about needs and attitudes regarding resilience training of undergraduate-medical-trainees in Middle-East-and-North-Africa-region. The purpose of this study is to investigate the value of an innovative curriculum developed through design-based-research to build resilience-skills among undergraduate-medical-trainees in the United-Arab-Emirates.

### Methods

Convergent-mixed-methods-study-design was utilized. Quantitative data collection was through controlled random group allocation conducted in one cohort of undergraduate medical students (n = 47). Students were randomly allocated into the respective resilience-skills-building-course (study-group) versus an unrelated curriculum (control-group). All students were tested at baseline (test-1), at end of 8-week course (test-2), and again 8 weeks after end of course (test-3). Then students crossed over to the opposite course and again tested at end of 8 weeks (test-4). Testing at four timepoints consisted of questionnaires related to *burnout*-Maslach-Burnout-Inventory; *anxiety*-General-Anxiety-Disorder-7; and *resilience*- Connor-Davidson-Resilience-Scale. Quantitative data were analysed descriptively

**Data availability statement:** Relevant data are within the manuscript and its Supporting Information files.

**Funding:** The author(s) received no specific funding for this work.

**Competing interests:** The authors have declared that no competing interests exist.

and inferentially. Qualitative data, constituting of students' perception of their experience with the course, was captured using virtual-focus-group-sessions. Qualitative analysis was inductive. Generated primary inferences were merged using joint-display-analysis.

## Results

Significant proportion of the students, at baseline, seemed to be at risk for burnout and anxiety, and would benefit from developing their resilience. There appeared to be no statistical differences in measures of burnout, anxiety, and resilience related to course delivery. Overall risk for anxiety among students increased following the COVID-19 lockdown. Qualitative analysis generated the 'Resilience-Skills'-Building-around-Undergraduate-Medical-Education-Transitions' conceptual model of five themes: Transitions, Adaptation, Added Value of course, Sustainability of effects of course, and Opportunities for improving course. Merging of findings led to a thorough understanding of how the resilience-skills'-building-course affected students' adaptability.

## Conclusion

This study indicates that a resilience-skills'-building-course may not instantly affect medical trainees' ratings of burnout, anxiety, and resilience. However, students likely engage with such an innovative course and its content to acquire and deploy skills to adapt to changes.

## Introduction

Burnout is a syndrome associated with long-term fatigue, physical exhaustion, hopelessness, and helplessness in people who are exposed to intense emotional demands due to their job, and who constantly engage with other people, with negative attitudes towards work, life, and/ or other people [1]. In simple terms, burnout can be defined as emotional exhaustion, depersonalization, and low personal accomplishment seen in individuals who have an intense relationship with people as part of their job [2–5]. Exhaustion, cynicism, and the emotional beliefs around lack of self-efficacy form the triadic symptoms of student burnout [6] with its prevalence higher among medical students relative to those enrolled in other disciplines [7]. The demanding medical curricula expect a higher academic investment, placing a heavy workload on the student leading to burnout [8], and giving rise to certain patterns of behaviour such as self-criticism and demandingness that could manifest in obsessive or self-sabotaging behaviours further aggravating the symptoms of burnout [9]. Medical students are also subjected to a chronically sustained stress that arises from caring for patients within a demanding and overloaded system that could lead to occupational burnout [10]. This is particularly true closer to graduation, and as resident doctors [11]. In fact, burnout or job-related distress, among medical trainees and healthcare professionals, is an increasingly recognized problem worldwide [12]. The emotional distress that permeates from burnout affects the medical students' education as well as their mental health with the potential of leading to debilitating consequences, including drop out from the medical programme and suicidal ideation [13]. Dropping out of university is associated with a range of consequences that could negatively affect not only the student but also the respective students' family, the faculty, and the wider community as it entails unmet expectations, frustration associated with the losses, including the financial ones [14].

Evidence indicates that burnout prevails and increases as students progress into their final years at medical school [15], and has been associated with high stress levels, depression, anxiety, substance abuse, and risk of suicide [16,17], all of which lead to suboptimal clinical performance and ultimately poor outcomes of care [18]. The deterioration of psychological wellbeing of medical students continuing as they become resident doctors [19] is a reality that can no longer be overlooked [20]. It prompted the creation of focused programmes to improve the mental health of students in medical school by focusing on reducing levels of depression, stress, and anxiety [21,22], and suicidal ideation [23], and by developing non-technical skills (complementary to those of basic and clinical medical sciences) required in the practice of medicine such as empathy and communication [24,25].

Universal efforts are needed to improve medical students' well-being [20,26]. Effective individual-focused interventions include mindfulness-based approaches [27–32], stress management and self-care training [33–35], communication skills training [36–39], and small group curricula [36,40,41]. It is worth anchoring any such learning opportunity in adult and experiential learning theories. These theories assume that adults are intrinsically motivated to learn, deploy self-regulated learning [42,43], and have mental models constructed from previous experiences that take the form of a growing resource for learning. These theories are also based on the premise that adult learners regularly exercise analogical reasoning in learning and practice [44]. This puts forward the Kolb's experiential learning cycle [45,46] which suggests that learning which occurs through concrete, hands-on experiences in a safe environment leads to reflective observation (where the learner identifies gaps in their mental models). Next, the learner adapts their mental constructs (i.e., abstract conceptualisation), then actively experiments using the adapted mental models in new experiences. Based on this constructivist perspective [47], the learning opportunity becomes a valuable resource for active experimentation. This fortifies new knowledge, and changes patterns of behaviours leading to long-term change in practices [45,47]. Among the previously pinpointed limitations of the Kolb's experiential learning theory is that it overlooks the fundamental learning that occurs in relating to others. The resultant simplistic view of experiential learning pulls it away from its origins, where it was sourced originally from human relations' training [48]. This is particularly relevant to the practice of medicine given its reliance on others, within the same health profession and interprofessionally, and also with patients and their families [49,50]. It is believed that cognition is a co-constituted process that is disseminated across the learner, the environment where learning occurs, and the activity that the learner is involved in [51]. The literature highlights the importance of conceptualizing experiential education in more sociological terms, illustrating how the individual learner is without doubt connected to environmental factors, including social and cultural ones, and is continuously interacting with others [52,53]. Hence, from a practical perspective when developing learning opportunities aimed at empowering medical students to improve their overall wellbeing, it is worth employing Kolb's experiential learning theory in conjunction with a social constructionism theory, where a small group of people learn through interacting among each other. Social constructivism proposes that knowledge is constructed through social processes and interactions [54,55]. This shows how the learner is actually embedded in the context of learning, and how participation and learning go together [56].

Anchoring educational endeavors in experiential learning models will enable the contribution to the theory around the subject matter along with providing practical value (i.e., improving the medical students' wellbeing). This is actually one of the key foundations of design-based research which is an approach that allows for iterative investigations of the learning intervention in an authentic educational context [57]. These investigations usually rely on mixed methods research design which enables the development of a thorough,

systemic understanding of the subject matter, where students become participants in the design and analysis [58,59]. Experts get to comment on an initial prototype of the learning intervention. This allows for iterative improvements of the initial design based on meaningful interactions between the involved researchers, designers, and/ or practitioners [60]. The developing prototype gets continuously evaluated by the involved stakeholders through interactive group meetings to distill lessons learned (in terms of the educational practice) and to advance the theories underlying the learning intervention (by virtue of its design). As such, design-based research constitutes a valuable opportunity to improve both the practice and theory around educational interventions [61].

Although curricula and programs teaching skills related to resilience and self-care are widely adopted, there is little knowledge related to the needs, attitudes, and acceptance of resilience training of undergraduate medical trainees in the Middle East and North Africa (MENA) region [62,63]. Also, although it is established that burnout and anxiety in medical students are widespread [64,65], yet due to the evident variation across countries, efforts to measure burnout in individual schools are warranted to better inform decisions around supporting the students and addressing their exact needs. Furthermore, specific interventions targeting resilience skills may be beneficial, where more research is needed to determine the most effective methods [66]. Accordingly, the overall purpose of this study is to investigate the value of an innovative curriculum developed, implemented, and evaluated in alignment with the principles of design-based research to build resilience skills among undergraduate medical students training in a multi-cultural, multi-ethnic urban setting in the United Arab Emirates (UAE). The respective curriculum is anchored in constructivism experiential learning theories and designed to foster self-directed learning. The current mixed methods research includes both quantitative and qualitative aspects. This research design was adapted for this study in order to answer the following three questions, corresponding [as per established mixed methods article reporting standards [67]] to the quantitative, qualitative, and integration components, respectively:

1. What are the baseline levels of burnout and emotional distress among medical students entering clinical training, and do these levels change over time and after exposure to a curricular course aimed at building the students' resilience?

2. How do medical students describe their lived experiences around the changes that are integral to their educational trajectory, and the resources that enabled them to effectively adapt (including but not limited to the respective resilience skills' building curriculum)?

3. How can a curricular course aimed at building students' resilience skills affect undergraduate medical students and their adaptability?

## Methods

### Context of the study

The current study was conducted at Mohammed Bin Rashid University of Medicine and Health Sciences (MBRU) in Dubai, UAE, with a single cohort of Year 4 medical students [68]. MBRU, through the College of medicine (CoM), offers a six-year Bachelor of Medicine and Bachelor of Surgery degree (MBBS) that is characterized by a spiral curriculum [69] made of three phases: foundational basic sciences, preclinical sciences, and clinical rotations. Phase 1, which corresponds to Year 1, serves as an introduction to fundamental medical concepts and basic human science. Phase 2, covering Years 2 and 3, focuses on the different body organ systems in relation to clinical medicine. Years 4, 5, and 6 constitute Phase 3. During Years 4

and 5, students undertake clinical placements in different private and public hospitals. During Year 6, students undertake an apprenticeship. The study cohort was comprised of 47 Year 4 medical students in the academic year 2019–2020 enrolled in the 'Resilience Skills' Building' course (that took place from 24th October 2019 through 16th April 2020). The cohort was from 17 nationalities (33 females and 14 males).

## Description of the resilience skills' building course

The resilience skills' building course was designed at the inception of the clinical rotations of the MBBS of MBRU in 2018, in alignment with the principles of design-based research [57,58]. This approach relies on design thinking, which is a process that deploys a combination of creativity and innovation to iteratively develop, implement, and evaluate a novel product [60]. Through this approach, the study investigators: a group of medical education practitioners, designers, and researchers, strove to attain practical impact in the educational practice in the context of this study, along with contributing to the advancement of knowledge in relation to the subject matter. The approach was design driven [57], and took place in authentic, real-life learning settings where the learning takes place normally, involving continuous cycles of design, evaluation, and redesign. Accordingly, the insights developed from evaluating the experience led to further improvement and redesign [61].

The course development process was initiated by a workshop aimed at discussing how best to build resilience among undergraduate medical students during their clinical training. The workshop was led by the chairperson of the clinical sciences department at CoM (S.B.H.) and its participants were diverse including faculty and staff members. To set the scene, S.B.H. presented a thorough review of the literature, highlighting medical students' susceptibility to mental health challenges and the importance of supporting them in proactively fostering their resilience. S.B.H. also showcased a thorough review of relevant curricula in medical schools, worldwide, and a selection of academic papers that focused on investigating the efficacy and/ or effectiveness of such learning and teaching interventions. The workshop participants were engaged in discussing and in turn agreeing on the content of the course and the means of its delivery.

This course was conceived as one of several 'longitudinal curricular themes' that are delivered alongside the clinical placements [70]. It was anchored in the theoretical principles of adult learning and Kolb's experiential learning, where reflection was fostered through embeddedness in the clinical environment, and the supervision of and feedback from the course instructors (including but not limited to a psychotherapist/ university counsellor). This underlying constructivist basis paved the way for enabling the students to assimilate novel information into preexisting knowledge, and for encouraging them to proactively learn by engaging with concrete experiences, integral to the course, and reflecting upon them. Within this context, from a social constructionism perspective, participation and learning go together, where the learner changes as a result of reflection on direct experiences.

The respective course is described extensively in a previously published study [70] which aimed at exploring the perception of the students regarding their understanding of, and personal experience with building resilience, and their engagement with the content of the course. In short, the course consists of 6 hours of instruction on building skills for resilience during the first clinical year of the MBBS (i.e., Year 4). The overall objective of this course is to raise awareness about the challenge of stress in the medical students' trajectory and the clinical workplace, and to provide tools for understanding, developing, and deploying resilience skills. The subjects covered as part of this course include: introduction to cognitive behavioural therapy, mental toughness, practicing mindfulness, emotional intelligence, coping strategies to increase personal resilience, and time management.

## Research design

A convergent mixed methods study design [71–74], which has been frequently commended in health professions' education research [69,75,76], was utilized to develop a thorough, systemic understanding of the effects of the resilience skills' building course on the participating undergraduate medical students (Fig 1). The adapted study design is characterized by three phases. In the first phase, quantitative and qualitative data were simultaneously collected. The quantitative and the qualitative data were analysed each independently, resulting in two sets of primary inferences, in the second phase of the study. In the third phase of the study, the primary inferences generated by the quantitative analysis were merged with that generated from the qualitative analysis, using the iterative joint display analysis process [77]. The integration of data types (i.e., quantitative and qualitative) is meant to raise the validity of the study's findings [78]. Among the reasons why such a thorough approach to research was selected for the current study is to fulfil one of the principles of design-based research that, as previously described, led the development, implementation, and evaluation of the 'resilience skills' building course'. Design-based research recommends deploying mixed methods approach to understand the underlying processes or factors [61].

Ethical approval for the study was granted by the Mohammed Bin Rashid University of Medicine and Health Sciences- Institutional Review Board (MBRU-IRB-2019-021).

## Data collection

**Quantitative.**  The quantitative data collection was through a controlled random group allocation, cross over design conducted in one cohort of 4th year undergraduate medical students starting their clinical training (n = 47). The only inclusion criterion was enrolment in the 4th year. All the quantitative data was collected between 24th October 2019 and 16th April 2020. Students were in clinical placements until 8th March 2020, after which they conducted their courses online. Students were randomly allocated (with gender balance) to taking the course on resilience skills (study group), and the other half was allocated to a different, unrelated course and considered the control group (Fig 2). The course, in which the students of the control group were enrolled in, is entitled: 'health system sciences', and was purposefully selected to be the one running in parallel to the course under investigation (i.e., resilience skills' building course/ longitudinal theme). The rationale of the selection is that, as per the requirement of such randomized controlled design [79], the respective 'health system sciences' course content, which revolves around acquiring performance improvement tools and

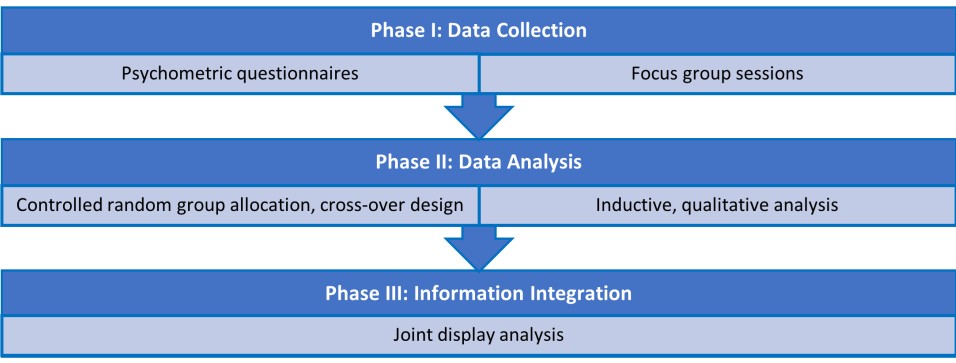

**Fig 1. Outline of the current study's convergent mixed methods research design.**

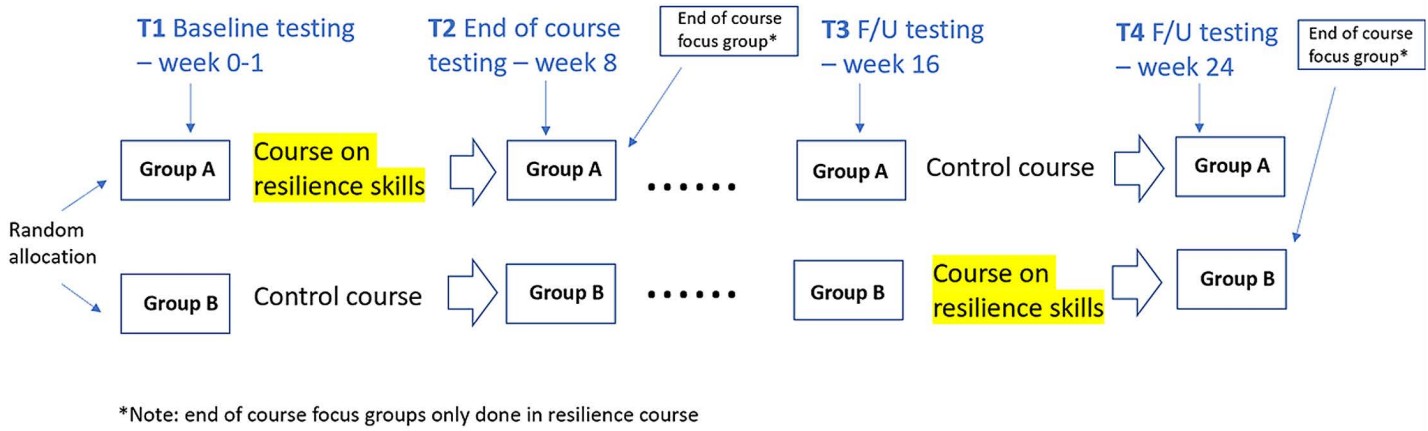

*Note: end of course focus groups only done in resilience course

**Fig 2. Outline of the controlled random group allocation, the cross over design, and the data collection.** All the quantitative data was collected between 24th October 2019 and 16th April 2020.

deploying them in healthcare settings, and its mode of delivery are completely different and not relevant to that of the course under investigation.

The students' participation in quantitative data collection was completely voluntary. Baseline testing was performed in study and control groups, after the students gave verbal informed consent to participate in the study (quantitative testing at four timepoints). At the beginning of the first session for each of the study and control groups, a neutral party (neither the study's principal investigator nor the course coordinator) read out loud a consent text that meets all established consent form for human subjects' research criteria [80–82]. The students were asked to complete the respective questionnaire only if they consent to participate in the corresponding research study. As such, completion of the questionnaires by the students constituted the evidence of their consent to participate in the study. The consent process and text (that was read out loud to the students) were approved by the abovementioned Institutional Review Board approval (MBRU-IRB-2019-021). Then, the study group received a resilience skills building course delivered over an 8-week period, whereas the control group received the unrelated curriculum during the same time. Both groups were tested again at the end of the curriculum period. After a 2-month interval, the groups were tested again and then cross over to the opposite curriculum, where the control group then received the resilience skills' building course. Testing was completed in both groups at the end of the second 8-week curriculum period. Testing at the four timepoints consisted of standardized questionnaires (Maslach Burnout Inventory, General Anxiety Disorder-7, and Connor-Davidson Resilience Scale).

## Instruments

*Maslach Burnout Inventory* (MBI). This is a widely used survey for workplace-related stress that has been validated in numerous populations, including healthcare workers and students in educational settings. It had been repetitively validated in MENA region, in general, and UAE, in specific [83–87]. The items evaluate the dimensions of emotional exhaustion (EX), cynicism (CY), and professional efficacy (PE). Subjects are asked to indicate their level of agreement with every item from a 7-point Likert-type scale. High scores on the EX and CY, and low scores on PE are indicative of the presence of work-stress or 'burnout' [88].

*Generalized Anxiety Disorder 7-item scale (GAD-7).* This well-recognized scale is validated internationally, and also in the MENA region, for measuring anxiety symptoms in clinical and research realms [89–91]. This happens through assigning scores of 0, 1, 2, and 3, to the response categories of 'not at all', 'several days', 'more than half the days', and 'nearly every day', respectively. GAD-7 total score for the seven items ranges from 0 to 21. Scores represent: 0–5 mild, 6–10 moderate, 1–15 moderately severe anxiety, and 5–21 severe anxiety. When used as a screening tool in healthcare settings, further evaluation is recommended when the score is 10 or greater [89]. A recent meta-analysis of the accuracy of the GAD-7 for identifying generalized anxiety disorder found that pooled sensitivity and specificity values appeared acceptable at a cutoff point of 8 [sensitivity: 0.83 (95% CI 0.71–0.91), specificity: 0.84 (95% CI 0.70–0.92)] although cutoff scores 7–10 also had similar pooled estimates of sensitivity/specificity [90].

*Connor-Davidson Resilience Scale (CD-RISC).* This scale is validated for measuring resilience. It had been validated in MENA region, in general, and UAE, in specific [92,93]. It comprises 25 items, each scored using a 5-point range of responses: not true at all (0), rarely true (1), sometimes true (2), often true (3), and true nearly all of the time (4). The subject scores the questions based on how they have felt over the past month. The total score ranges from 0–100, and higher scores reflect greater resilience [94].

**Qualitative.** The participating students were invited to externalize their thoughts in relation to their lived experiences with the changes that are integral to their educational trajectory, and the resources that enabled them to effectively adapt (including but not limited to the resilience skills' building course). The perception of students was captured using two online focus group sessions, one corresponding to each of the two groups of students (Fig 2). As a qualitative data collection tool, focus group was selected for the current study since it is informative, allowing for probing and rich discussions [95]. Besides discussing their preexisting ideas, participants in these focus groups sessions get to reflect upon and provide feedback on new information mentioned by the other participants. The facilitator gets to delve into intricate topics, attitudes, and lived experiences that may prove to be challenging to capture through other techniques, be it quantitative or qualitative [96].

Both focus group sessions took place 1st and 15th May 2020. For each session, 8 randomly selected students (from the corresponding group) were invited to a one-hour, virtual focus group session that followed a preset protocol (Appendix I), composed of four segments. The first 10 minutes of each session focused on creating the space for the students to reflect on their personal journey since the beginning of the clinical phase, and then the following 10 minutes led the students towards sharing how they perceive their personal and collective evolution throughout the respective phase. Then, the facilitator anchored a 15-minute discussion around the resilience skills' building course which enabled the students to discuss how they perceive the course affected them. The last 15 minutes of the focus group sessions constituted of a macro-level reflection on the overall experience of the course and the acquired competencies. Prior to conducting the focus group sessions, the protocol underwent face and content validation (by four faculty members, two of whom have key leadership roles in the planning, delivery, and evaluation of the clinical phase).

The students' participation in this focus group sessions was completely voluntary, and each participant was required to provide verbal informed consent prior to the commencement of the sessions. At the beginning of each session, an investigator other than the respective session facilitator read out loud a consent text that meets all established consent form for human subjects' research criteria [80–82]. The students were asked to stay online to engage in the conversation only if they consent to participate in the corresponding research study. As such, choosing to stay online constituted the evidence of their consent to participate in the study. The consent process and text (that was read out loud to the students) were approved by the abovementioned Institutional Review Board approval (MBRU-IRB-2019-021). The sessions

were recorded. They were both facilitated by the same researcher, who is trained in socio-behavioural research (F.O.). Another member of the research team (B.N.), who is trained in understanding the nuances of human communication and behaviour, attended both sessions for the primary purpose of notetaking.

To protect the anonymity of the participants, each was assigned a unique identifier, composed of two parts: a serial number (i.e., 01 to 11), followed by 'F' for female or 'M' for male. For example, the identifier: 5F, represents participant number 5, who is a female.

## Data analysis

**Quantitative.** The quantitative data collected from the testing at the four timepoints were analysed using SPSS software version 24, descriptively and inferentially. All the continuous data were tested for normality using the Shapiro-Wilk test. Continuous data was described by using measure of tendency and dispersion. Longitudinal data was analysed using Multivariate Repeated Measures ANOVA (MRANOVA) to assess the effects of independent variables on multiple dependent factors. For this analysis, we had three factors: two groups (A and B), subject types (MBI EX, MBI CY, MBI PE), and time. ANOVA was used to compare the means over different times for GAD7 and CDRI data. Following the MRANOVA and ANOVA, post hoc analyses were used to explore specific pairwise comparisons between times. T-tests were used to compare means between the two groups for normally distributed data. P-value less than 0.05 was considered significant for all the tests.

**Qualitative.** The qualitative data analysis started after the conclusion of the data collection phase. The iterative data analysis was inductive [97]. The analysis process was based on constructivist epistemology [98]. This was done, by two data analysers (F.O. and B.N.), using a participant-focused approach to phenomenological thematic analysis. Prior to the analysis, they pointed out personal characteristics that they believed could affect their perceptions in relation to the subject matter. Consistency, regarding the underlying assumptions and theories, was maintained throughout the process by one of the two data analysers (F.O.). By embracing rather than avoiding the investigators' personal involvement in the research and by evaluating interpretations according to their impact on participants, investigators, and readers [99], the quality control deployed in this investigation shifted from the objective truth of statements to understanding by people. This unique interpretative approach involves the ability to recognize and recreate the experiences of the participants. The purpose of this approach is to understand and relate to individual participants, and their attitudes and behaviours, rather than to find casual explanations. This methodology assumes that trained qualitative researchers can interpret individuals' emotions, thoughts, and behaviours by actively listening and comprehending what their self-expressions.

The qualitative analysis process followed the Braun and Clarke six-step framework [98]. This multi-phased approach to thematic analysis has been widely deployed in research around health professions' education [100]. NVivo software version 12.0 plus (QSR International Pty. Ltd., Chadstone, Australia) enabled coding the data (i.e., assign labels/ titles to the sub-categories, categories, and themes), and in turn classifying the text fragments tagged by the data analysers.

The analysis process involved the following steps:

1. Familiarization with the dataset.

The data analysers took turns in reading out loud the essays to familiarize themselves with the compiled dataset. They thoroughly reflected upon the content of the de-identified data, and shared any thoughts that surfaced for them in this regard.

2. Development of initial codes.

Segments of text from the focus group transcripts that related, directly or indirectly, to the overall purpose of the study were extracted. Accordingly, any text fragment relating to the participants' understanding of lived experiences around the changes that are integral to their educational trajectory, and the resources that enabled them to effectively adapt (including but not limited to the resilience skills' building course) was tagged.

This process continued until data saturation was achieved. This systematic review led to the formation of categories of text fragments which lent itself to the next step.

3. Searching for themes

This step included carrying-out several rounds of structured reflections, through which the different ways by which the identified categories could relate to one another were identified (leading to several potential interconnections).

4. Reviewing themes

The categories were collated to form higher-order themes, according to the linkages that made most sense to the two data analysers (Fig 3).

5. Coding and defining categories and themes.

All the categories and themes were then given labels (i.e., coded) and defined in the context of the study. This resulted in the study's conceptual model.

After the completion of the fifth step and prior moving to the last step of the adapted analysis framework, a respondent validation was conducted, where a random sample of the study's participating students were invited to attend a virtual meeting. The faculty member who delivered most of the resilience skills' building course content (B.N.) showcased the research question corresponding to the qualitative component of the current study, the multi-phased process of qualitative analysis, and the generated conceptual model. After reflecting upon the perceived meaning of the generated conceptual model, the meeting attendees agreed with all the identified codes (i.e., themes and categories), and how the current study's conceptual model depicts the linkages across them.

6. Reporting on findings

The results of the abovementioned steps were reported narratively in alignment with established guidelines, including the Standards for Reporting Qualitative Research (SRQR) [67,101,102] as part of the framework's sixth step (i.e., Results section). This narrative reporting on qualitative results involves the provision of examples of the text fragments that got encapsulated in the corresponding categories. The broader observations (i.e., primary inferences) are described in the narration around the fragments that were selected. The examples are not supposed to fully represent the preceding narration, they are meant to convey the 'essence' only in order to reinforce the narration. As such, this reporting on qualitative results technique is not about interpreting the selected exemplars, but rather about reflecting on all the fragments encapsulated in the respective categories and using the exemplars to build on the inductively generated storyline.

To further substantiate the findings, a tally was conducted to record the frequency of text fragments within each category, within the identified themes. If for a single participant, more than one relevant text fragment was put in the same category, they were all collectively considered as one entry. Thus, the tally reflected the number of participants that brought up matters relevant to each of the categories. For each theme, the numbers of participants corresponding to the encapsulated categories were summed, and the ratio of females to males entries was determined.

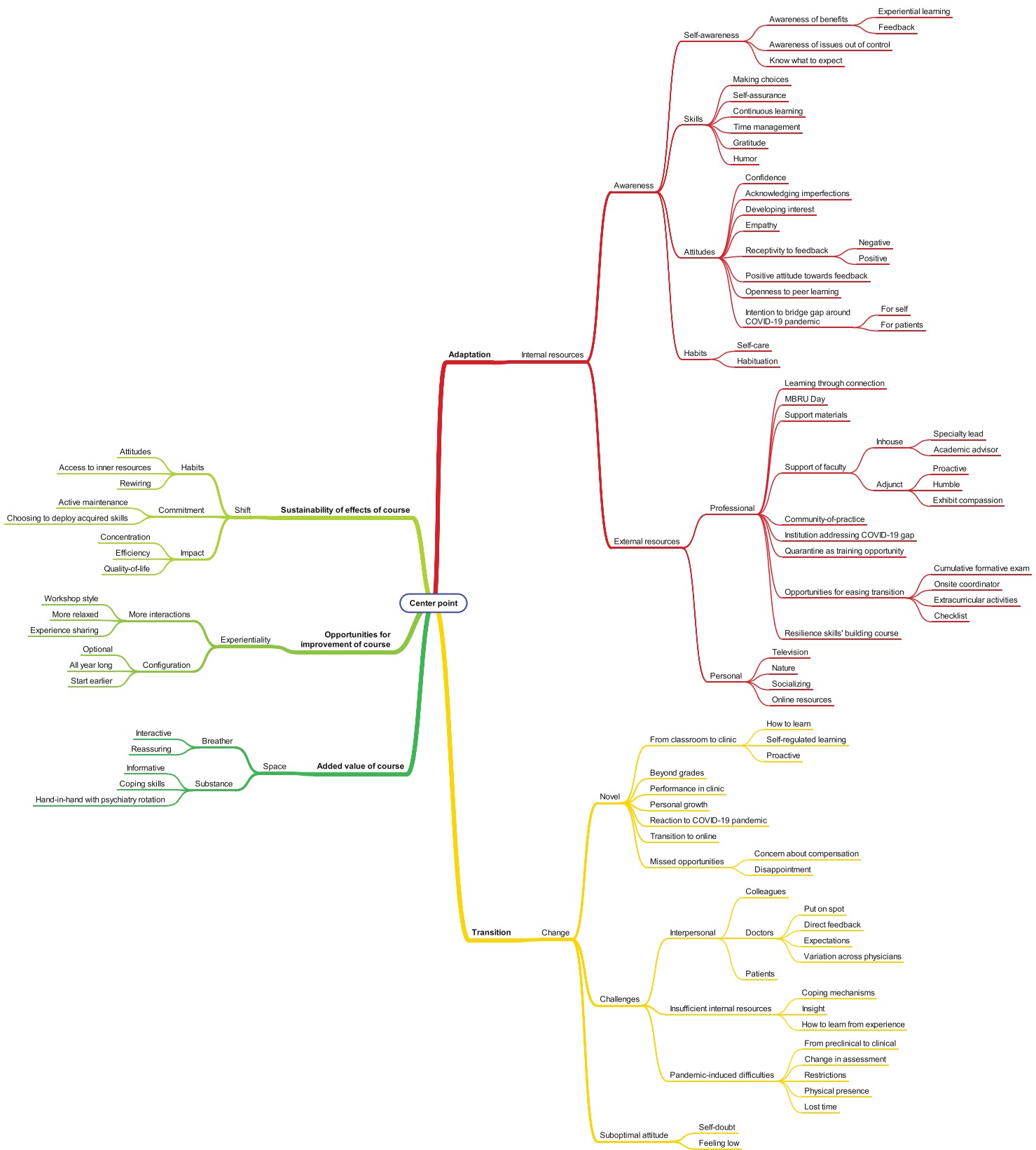

**Fig 3. Mind map deployed as a tool to facilitate the qualitative analysis.**

### Integration

After the completion of the independent data analyses of quantitative and qualitative data, the generated primary inferences were merged using the iterative joint display analysis process [103]. This stage allowed for drawing meta-inferences from the mapping of findings generated from each of the independent preceding analyses (Fig 1) [104]. As such, the researchers were able to identify where the primary inferences build upon (or at least confirm) each other. This systemic method of analysis also enabled identifying findings that complement each other.

## Results

In alignment with the guidelines of reporting on mixed methods research [67] that were adhered to for the current study, the analysis of the quantitative data addressed the first research question of the current study, while the output of qualitative analysis answered the second research question. Furthermore, as previously mentioned in the Methodology section, the third research question was addressed through the integration of quantitative and qualitative findings.

### Output of quantitative analysis

The analysis revealed changes in questionnaire scores related to burnout (Table 1), anxiety, and resilience over time (Table 2).

At baseline, the percentage of students with high risk for burnout (MBI scores EX > 18, CY > 9, and PE < 33) was 42.4% (Fig 4 and Fig 5) and high risk for anxiety (GAD score ≥ 10) was 34% (Fig 6). Baseline resilience, as measured by CDRS, was 67.9 (±17.3 SD) (Fig 7).

No statistical differences were noted in these scales at the end of the course (test 2) or at the two-month follow-up timepoint (test 3). Crossover occurred just prior to the world-wide COVID-19 pandemic.

Since there were no between-group differences, the scores of all tests were summated for the entire group and followed over the four time periods. Overall, the number of students

**Table 1. MBI questionnaire results over time periods T1–T4 [Mean +/− Standard Error of Mean (SEM)].**

| Group | MBI EX | | | | MBI CY | | | | MBI PE | | | |
|---|---|---|---|---|---|---|---|---|---|---|---|---|
| | T1 | T2 | T3 | T4 | T1 | T2 | T3 | T4 | T1 | T2 | T3 | T4 |
| Group A | 3.20 ± 0.46 | 3.29 ± 0.46 | 3.24 ± 0.37 | 2.89 ± 0.61 | 2.46 ± 0.46 | 2.86 ± 0.58 | 2.66 ± 0.29 | 2.51 ± 0.66 | 3.58 ± 0.38 | 3.82 ± 0.48 | 4.27 ± 0.18 | 4.45 ± 0.42 |
| n = | 16 | 11 | 20 | 7 | 16 | 11 | 20 | 7 | 16 | 11 | 20 | 7 |
| Group B | 3.83 ± 0.32 | 3.40 ± 0.30 | 3.80 ± 0.32 | 4.06 ± 0.40 | 2.58 ± 0.24 | 2.25 ± 0.38 | 2.89 ± 0.32 | 2.76 ± 0.43 | 3.42 ± 0.37 | 4.04 ± 0.25 | 3.87 ± 0.29 | 4.02 ± 0.13 |
| n = | 18 | 11 | 20 | 10 | 18 | 11 | 20 | 10 | 18 | 11 | 20 | 10 |
| Total all students | 3.54 ± 0.28 | 3.35 ± 0.28 | 3.52 ± 0.25 | 3.58 ± 0.36 | 2.52 ± 0.25 | 2.55 ± 0.35 | 2.78 ± 0.22 | 2.66 ± 0.36 | 3.49 ± 0.25 | 3.93 ± 0.26 | 4.07 ± 0.17 | 4.20 ± 0.19* |
| n = | 34 | 34 | 40 | 17 | 34 | 22 | 40 | 17 | 34 | 22 | 40 | 17 |

T: Timepoint.

MBI: Maslach Burnout Inventory; MBI EX: MBI-Exhaustion (higher score worse); MBI CY: MBI-Cynicism (higher score worse); MBI PE: MBI-Professional Efficacy (higher score better).

Gray shading: Delivery of resilience skills' building course – Between T1 and T2 for Group A and between T3 and T4 for Group B.

ANOVA with repeated measures analysis – no differences between groups over time.

Post-hoc analysis:

*p = 0.036 vs T1 two-sided t-test.

**p=0.054 vs T1 two-sided t-test

**Table 2. GAD and CD-RISC questionnaire results over time periods T1–T4 (Mean +/− SEM).**

| Group | GAD7 | | | | CD RISC | | | |
|---|---|---|---|---|---|---|---|---|
| | T1 | T2 | T3 | T4 | T1 | T2 | T3 | T4 |
| **Group A** | 7.88 ± 1.47 | 8.50 ± 1.87 | 7.00 ± 1.15 | 10.86 ± 2.26 | 69.16 ± 3.71 | 70.58 ± 3.88 | 64.15 ± 3.30 | 71.86 ± 4.85 |
| n = | 16 | 10 | 20 | 7 | 19 | 12 | 20 | 7 |
| **Group B** | 6.69 ± 1.08 | 8.73 ± 1.41 | 8.40 ± 1.46 | 9.90 ± 1.39 | 66.63 ± 4.28 | 71.00 ± 3.01 | 62.50 ± 3.80 | 68.30 ± 3.52 |
| n = | 16 | 11 | 20 | 10 | 19 | 11 | 20 | 10 |
| **Total all students** | 7.28 ± 0.91 | 8.62 ± 1.13 | 7.70 ± 0.92 | 10.29 ± 1.20** | 67.89 ± 2.80 | 70.78 ± 2.43 | 63.33 ± 2.49 | 69.76 ± 2.81 |
| n = | 32 | 21 | 40 | 17 | 32 | 23 | 40 | 17 |

T: Timepoint.

Gray shading: Delivery of resilience skills' building course – Between T1 and T2 for Group A and between T3 and T4 for Group B.

ANOVA with repeated measures analysis – no differences between groups over time.

Post-hoc analysis:

*p = 0.036 vs T1 two-sided t-test.

**p=0.054 vs T1 two-sided t-test

GAD7: Generalized Anxiety Disorder-7 scale; CD RISC: Connor-Davidson Resilience Scale

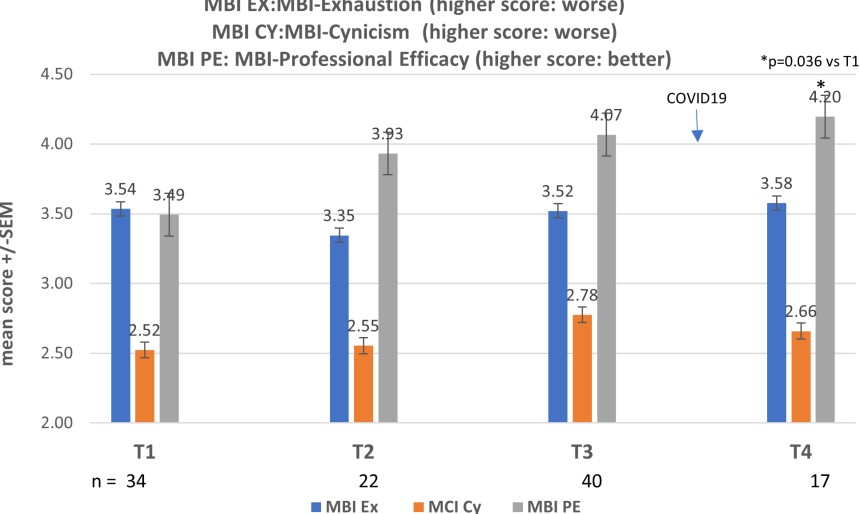

**Fig 4. MBI results over 4 timepoints.**

with high risk for anxiety (GAD score ≥ 10) increased from 34% at baseline to 53% at the last timepoint (test 4), which was during the COVID-19 pandemic.

For the MBI, there appeared to be trend for a steady improvement in PE over time (Fig 4).

## Output of qualitative analysis

Out of the 16 students invited to both focus group sessions, 11 participated (5 from the first group and 6 from the second group, and 6 females and 5 males).

The qualitative analysis generated, as per this study's conceptual model: 'Resilience Skills' Building around Undergraduate Medical Education Transitions' (Fig 8), five interlinked themes namely: Transitions, Adaptation, Added Value of course, Sustainability of effects of course, and Opportunities for improving course. While the first two themes are generic. The latter three are

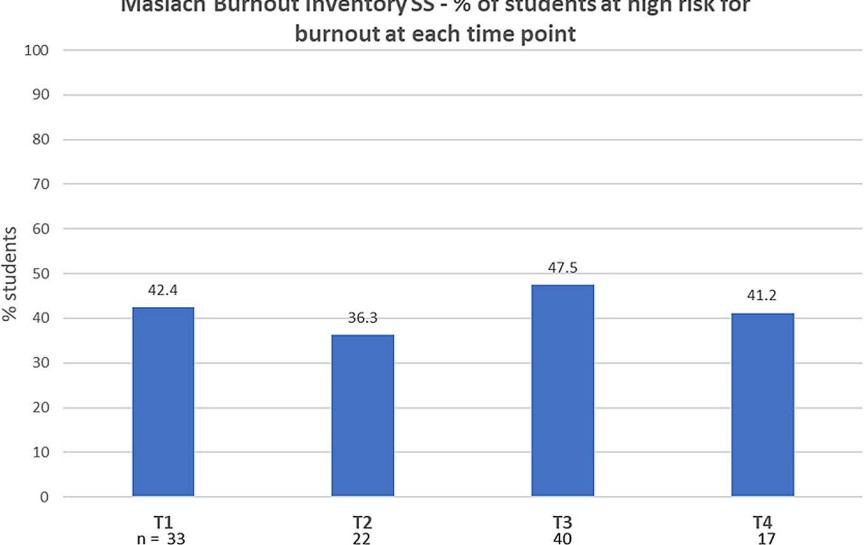

**Fig 5. Percentage of students at high risk for burnout at 4 timepoints (with no statistically significant difference across the 4 timepoints).** SS: Student Survey.

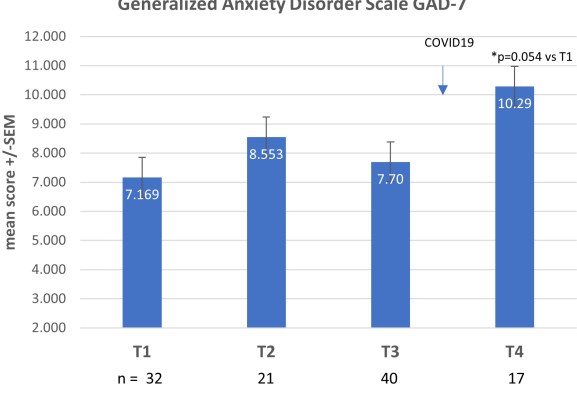

**Fig 6. GAD-7 results over 4 timepoints.**

specific to the resilience skills' building course under investigation in the current study. Within the Transitions' theme, three categories are identified: Changes, Challenges, and Suboptimal attitudes. As for the Adaptation theme, it includes two categories: Internal resources and External resources. Within the Added value of the course theme, there are two categories: Space and Substance. In relation to the Sustainability of effects of course theme, three categories are specified: Shift, Commitment, and Impact. Lastly, within the Opportunities for improving course theme, the following categories were identified: Experientiality and Configuration.

The tally of the count of text fragments belonging to each category showed the distribution, outlined in Table 3.

## Transitions

This theme encapsulates fragments of texts that refer to the learners' perception of the noticeable changes that they had been going through, and the associated internal and external experiences.

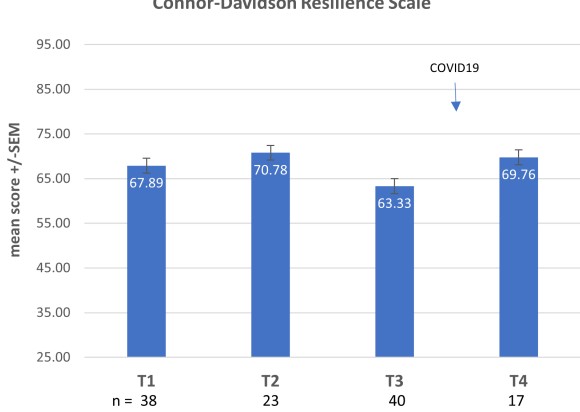

**Fig 7. CDRS results over 4 timepoints.**

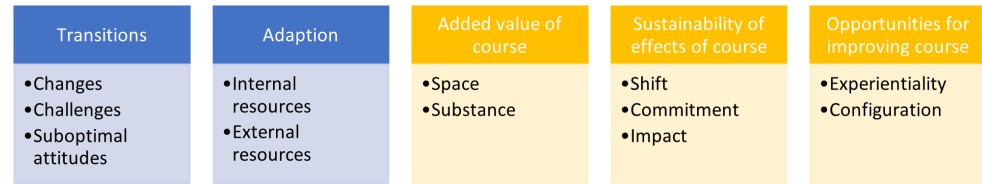

**Fig 8. The study's conceptual model: 'Resilience Skills' Building around Undergraduate Medical Education Transitions' (Blue themes are generic and Yellow ones are specific to the resilience skills' building course under investigation).**

**Table 3. Semi-quantitative tally of the output of the participant-focused qualitative analysis (Blue themes are generic and Yellow ones are specific to the resilience skills' building course under investigation).**

| Theme | Transitions | | | Adaptation | | Added value of course | | Sustainability of effects of course | | | Opportunities for improving course | |
|---|---|---|---|---|---|---|---|---|---|---|---|---|
| Category | Changes | Chal-lenges | Suboptimal attitudes | Internal resources | External resources | Space | Sub-stance | Shift | Commit-ment | Impact | Experi-entiality | Config-uration |
| Tally (n) | 4 | 7 | 6 | 7 | 10 | 7 | 4 | 5 | 4 | 2 | 5 | 6 |
| Sum (n) | 17 | | | 17 | | 11 | | 11 | | | 11 | |
| f:m ratio | 12:5 | | | 11:6 | | 6:5 | | 7:4 | | | 7:4 | |

n = number of individual participants, f = female, m = male

**Changes.** This category refers to the learners' reflections around noticeable shifts in external variables (expected or unexpected), and their perception of what they believe was required 'to rise to the occasion'.

> 1M: "…it was really new…we were previously always in lectures in classrooms and then we had to put ourselves out there…"

It taps into the learners' perception of the entailed novelty during times of change, and their firsthand realization of unpredictability and uncertainty.

> *3M: "…COVID-19 was definitely something new, unpredictable. I felt stressed because we, at first, did not know how to react to the changes accompanying the pandemic; we did not know what to expect…"*

Fragments of text around learners' descriptions of the expected transition from basic to clinical medical sciences' learning environment are encapsulated in this category. In times of transition, it became apparent to the learners that they needed to be more proactive, employing self-regulated learning, taking on more responsibilities.

> *4F: "…for me, the difference was around 'how to learn'. I was so used to having the material in front of me… When you are in the hospital, you sort of need to seek the information on your own whether that is asking doctors, getting laboratory results… you find yourself in the Operating Theatre for a surgery, for example, you need to learn differently, to go out of your way to learn…"*

Apparently, these changes led them to think beyond grades, recognizing the importance of their performance in the clinic.

> *2F: "…it was no longer about 'a grade' or how you performed in an exam…"*

The learners described their personal growth beyond academics.

> *2F: "…first day of Phase III compared to where I stand now, I think there has been a big change not only academically, as in increase in knowledge, but also personally… I learned and grew so much through the clinical experiences in terms of my personality…"*

Besides their description of transitioning from basic to clinical medical sciences, the learners also tapped into the unexpected changes due to the pandemic. They described the reaction of transitioning to distance learning via the online environment.

> *3M: "…I did not know what to expect from the online learning experience and how, in particular, we will compensate for not going to the hospital…"*

They also reflected upon their disappointment about missing out, and concerns around compensating for the entailed missed opportunities.

> *3M: "…I was looking forward to my surgery rotation, a lot of people had told me that it is really nice that you are going to have so much fun, and then, when the COVID-19 happened, we could not go and attend the rotation… I was disappointed because I really wanted to experience the rotation…"*

**Challenges.** This category refers to the fragments of text that shed light on the hurdles that surfaced for the learners while they were experiencing the externally triggered changes.

Some of the difficulties that they had faced were around interpersonal relationships, where they repetitively alluded to communication challenges. The learners touched upon the challenge of finding themselves in situations among their colleagues, with differing points of view.

> *11F: "…one of the biggest challenges I had to face was communicating with my colleagues who had different viewpoints. A lot of times, we seemed to be on different ends of*

*a spectrum. I would spend my day worrying about what this person said to me without actually learning…"*

They also mentioned several challenging aspects of their relationship with the supervising physicians, where for example, the learners felt like they were 'put on the spot'.

*1M: "…doctors would actually ask us questions, and sometimes put us 'on the spot'…"*

Some learners were yet to get used to receiving feedback directly from the physicians.

*2F: "…you find yourself right in front of the doctor, you get direct feedback, then and there…"*

Other learners brought-up feeling confused due to variation across physicians, and in some cases, unclear (or even high) expectations.

*5F: "…the change of assessment from preclinical to clinical brought me a lot of stress… variation in the way the assessors would approach the assessment form. I understand every doctor has a different style, as they graduate from different universities and have different training experiences, but when it comes to assessments, one would assume that it is standard across all rotations and all the doctors…"*

*9F: "…sometimes it would be challenging, during clinical rotations, to keep-up with the studying, as per the doctors' expectations…"*

Within the interpersonal realm, the learners also touched upon the novelty of needing to interact with patients.

*1M: "…we had to see and interact with patients…"*

According to the learners, they felt they had insufficient internal resources during times of change. This included inadequate coping mechanisms, realization of limited insight, and challenges around figuring-out how to learn from what they were experiencing.

*6F: "…the students do not really know what to do or how to do what they need to do to benefit from the clinical rotation… it may be useful to train the students about how to maximize their learning in the clinic…"*

They also alluded to difficulties associated with moving from preclinical to clinical phases, where they reflected on changes in assessments, more restrictions, and requirement of physical presence.

*5F: "…this year has been challenging on many levels…the change of assessment from pre-clinical to clinical brought me a lot of stress…"*

They also reflected on the lost or suboptimal utilization of time.

*6F: "…a lot of the students do not know how to benefit from their time in the hospital… we really are not used to the physical load that clinics put on us…"*

*9F: "…there is a lot of time that tends to be lost in transportation, trying to sleep early so one can wake-up early. It can get really messy…"*

The learners also mentioned pandemic-induced difficulties, including the restrictions in general and those around physical presence in particular, and the corresponding lost time, along with the modifications to the assessments.

> *3M: "…COVID-19 would count as a turning point which made our journey more challenging…"*

**Suboptimal attitudes.** This category encapsulates the fragments of text that shed light on the learners' reflections on their initial resistance to the changes and on the generated insecurities.

The learners reflected upon feelings of inadequacy.

> *3M: "…as a consequence to COVID-19 and missing-out on the clinical learning experience, the question that kept coming to my mind was: 'what if I am not capable and/ or confident enough to take care of patients?'…"*

> *11F: "…I keep getting this feeling of inadequacy, in general. You just feel that you are not good enough, whether you are with a consultant or a student…"*

It seems they were, at the times of transition, experiencing plenty of self-doubt, along with low confidence and limited self-efficacy.

> *5F: "…at the beginning of the clerkship, I had a lot of self-doubt. I was not sure if I knew enough and if what I knew was correct..,"*

They also referred to their mood during times of change.

> *9F: "… I was scared…my confidence did taper down in the past few months, with the quarantine and so, it took a toll on me when everything went online. I do not think my coping mechanisms were enough for me to keep-up with that change…"*

## Adaptation

This theme encapsulates fragments of text that refer to the learners' descriptions of how they manoeuvred through the difficulties associated with the changes. They reflected upon the coping strategies that they had deployed.

**Internal resources.** This category refers to the text segments that signify the intrinsic resources that the learners were able to deploy (whether consciously or subconsciously) as part of the adaptation process.

> *2F: "…I developed a 'thicker skin…"*

> *5F: "…I slowly found myself more drawn to the clinical aspect of things. I found myself staying at the hospital for a longer duration. Somehow my genuine interest and engagement were validating…"*

The learners referred to their awareness and knowledge base as enabling factors. They reflected upon self-awareness, and their acknowledgement of the benefits of experiential learning and feedback.

> *1M: "…It was a new kind of experience that we had to adapt to. …"*

*2F: "…These complementary skills cannot be learned through lectures and books, only… matters were no longer 'black-and-white…"*

They seemed to believe that their awareness of what issues are out of control also enabled them throughout the process.

*4F: "…it helped to remain conscious that COVID-19 was not in anyone's control. Also, nobody could have predicted it…"*

The learners also suggested that early curricular clinical exposure helped them, through managing their expectations.

*6F: "…transition to Phase 3 was tough, but the good thing for me was that I had a little bit of previous experience in the clinic, as part of a course in Phase 2; I knew what to expect…"*

The learners also identified skills that they believed helped them in getting through. They reflected on making choices, such as distancing oneself from others.

*11F: "…the only coping mechanism that worked for me was choosing to distance myself from people who did not make me feel so great and approach doctors instead…"*

They also brought-up continuous learning, exercising gratitude, humour, and self-assurance.

*1M: "…I learned a lot in the transition to phase III and to that I am so grateful…"*

*4F: "…humour also helped me; I really like joking about my situation, and after going through the psychiatry module, I learnt that humour was a mature coping mechanism…"*

Time management was also brought-up.

*9F: "…my capacity to effectively manage my time was built as I progressed across the rotations. I realized the studying needs and doctors' expectations for each rotation are different, so I kept coming-up with different schedules. I had to make sure that not only am I covering adequate cases, I am doing my readings and my own studying…"*

The learners also alluded to proactive adjustments in attitudes, where confidence, acknowledging imperfections, and allowing oneself to make mistakes were pointed out.

*4 F: "…I am willing to work towards continuously developing myself as a doctor for the rest of my life…I have internalized that medicine as a practice is a work-in-progress, I am never going to be perfect…"*

*9F: "…later my confidence built-up. I would talk to doctors, even if I said a wrong answer or if I have concern that my question could be considered 'stupid'. I just would shoot my shot, and I think that was good…"*

The learners also appreciated that their growing interest in the subject of the rotation also enabled them in effectively addressing the entailed challenges.

> *5 F: "…I found myself staying at the hospital for a longer duration. Somehow my genuine interest and engagement was validating. My self-doubt was still there. Yet, I started to feel that I am doing okay, that I am doing something right. I learned to love what I was doing, the hospital setting, the clinical environment…"*

They also discussed empathy.

> *2F: "…I became more aware of social cues and more empathetic…"*

They frequently alluded to how their relationship with feedback evolved. They realized that they became more receptive to it (negative and positive), which apparently, came hand-in-hand with a positive shift in attitude towards it.

> *2F: "…I became more receptive to feedback, whether it is negative (i.e., criticism) or positive…to acknowledge the importance of feedback and to use it to grow. In other words, to have a positive mindset towards feedback…"*

Apparently, becoming more open to peer learning was among the gradual changes that they identified as a catalyst, along with their intention to bridge the gaps that were generated around COVID-19, be it for themselves or for the patients that they were dealing with.

> *3M: "…just learning from the experiences of my peers… I am trying my best to see how I can make up for the missed clinical experience due to COVID-19, not only for myself also for patients' sake…"*

The learners also reflected upon habits that they believe helped them in manoeuvring through the challenging times. These habits included self-care, referring to allowing oneself to rest and take breaks.

> *4F: "…Also, the insight of when to rest and when to take breaks and that it is okay to take breaks was another good coping mechanism for me…"*

They also reflected upon how choosing to do something and doing it consistently until it becomes a habit is an enabling force in times of change.

> *11F: "…It is unlikely to be competent in anything when you are doing it for the first time… I do not know if it is a healthy coping mechanism, but I got to a point where I would actually stay back at the hospital until midnight to work and it was a great learning experience…"*

**External resources.** This category includes the text segments that refer to the extrinsic resources which the learners were able to access and deploy (whether consciously or subconsciously) to facilitate the adaptation process.

These resources include personal aspects, along with professional, program-related ones. The personal aspects include watching television, socializing, and deploying readily available online resources which support the learners' mental health.

> *1M: "…I am a 'television shows/movies' person so honestly my go to, whenever I am stressed or something, is always to watch an episode of a show/ series or a movie, or maybe go out with my friends to refresh and then go back to studying…"*

> *3M: "…one of the coping strategies that I usually use is to talk with my friends, or go out with them and discuss matters; I sometimes do that with friends who are not from the university so I get a different perspective as to the matters I may be dealing with or issues that I may have…"*

The learners also mentioned nature as a valuable external resource.

> *11F "…one biggest method which actually worked for me was basically going out into nature…The best way for me was to go outdoors and walk on the track alone, reflecting on how my day went and that alone I think had the biggest impact on me… it greatly helped me during my rotations…"*

In terms of the professional resources, the leaners reflected upon learning through connection with peers who have prior experience.

> *1M: "…a lot of the students helped me especially given that we were doing different rotations at the same time. So, for example, I started with psychiatry and after that I had family medicine so I used to speak to the students who finished family medicine and they would give me so many tips about where to go, how to do matters, how to study, what resources to use…"*

> *2F: "…I am really grateful for our batch and how much we have helped each other…I always had the option to go back to my colleagues, they would reassure me, and be like: 'it is not unexpected to feel this way in this particular rotation' or they may give me heads-up: 'in this scenario, expect to feel/ think this way'. They would shed light on study resources to enable me to find my own way; self-directed learning. I am grateful for the fact that I had guidance, and that we all helped each other. I think that that was a huge enabling factor for me this year…"*

The learners repeatedly reflected upon how the learners were helping each other, where their sense of togetherness and cohesiveness contributed to their resilience in times of change.

> *3M: "…our batch have been very co-operative and very helpful in sharing their experiences which helped us in managing our own expectations in the upcoming rotations…"*

The program-related external resources included the MBRU Day, where the students get didactic curricular sessions on Thursday of every week. These sessions are complementary to their clinical rotations.

> *1M: "…family medicine weekly sessions on Thursdays were very useful on many levels… they were really helpful not only for the rotation, but for the exams. The way these team-based learning sessions were structured helped in maximizing the learning…"*

The learners also brought-up the support studying material that they were offered, along with the support of select faculty. Some of those faculty members are inhouse, such as the specialty lead and academic advisor.

> *1M: "…I sincerely appreciate the family medicine lead; a very nurturing human-being…also, she always used to assign to us useful readings…"*

Other faculty members were adjunct ones, who exhibited proactiveness and humbleness.

> *2F: "…I really appreciate when doctors go the extra mile to teach us beyond what we see every day in the hospital…"*

> *3M: "……a number of physicians were really friendly; the way they talked to us was not authoritative… for example, some regularly shared their opinions or how their professional lives, after medical school, turned-out to be…hearing, from different people, what they went through widened my horizons, encouraged me to reflect more on what I want to do…"*

The professionalism of the adjunct faculty, including their compassion and empathy, was also brought-up.

> *5F: "…. I am so thankful, though, that I had very good, nurturing doctors. They taught us with so much compassion and empathy. This was enabling despite my self-doubt…so one thing that was helpful for me was the meeting with my academic advisor which helped me destress as I was able to have an open discussion in a very safe environment…"*

The learners also expressed appreciation to the feelings that they have of belonging to a community-of-practice, and to the sense of togetherness that they sensed in how the institution addressed the gaps due to COVID-19. A few learners implied that the quarantine was in a way a training opportunity.

> *4F: "…our batch was giving, and there was a strong sense of togetherness when it came to every single rotation. So that really helped us because we could rely on each other even though practically each of us had to do the work alone… the institution, as a whole, was taking steps to make sure our education was still at par of what they had intended prior to the onset of the pandemic…"*

The learners also indicated, in their reflections, some opportunities for easing the transition into the clinical environment of following cohorts. These included assigning an onsite coordinator and carrying-out a cumulative formative assessment.

> *6F: "…something that I think would have been very helpful to us in the rotation is having a faculty member onsite (present with us, in person), who is well informed about the hospital staff and us, to resort to when needed then and there… someone who knows how the hospitalize runs, who knows the doctors, who knows our schedules, someone who is 'in between' to facilitate…"*

> *7M: "…I want to suggest having students go through a 'transition exam' between phase 2 and phase 3 that covers the organ systems mainly. Just a pass/ fail exam. If the students pass this exam, it will reassure them that we are ready to start year 4/ phase 3.…"*

The learners also mentioned organizing for extracurricular activities.

> *9F: "…we have a hard time trying to fit in extracurriculars however, if we can somehow have these more fitted in our calendar (e.g., have trips or extracurricular activities, that would help us destress)…"*

They also suggested having a checklist of minimal requirements that a learner needs to perform in the clinical environment.

*10M: "…I think it would be helpful if we are provided with some sort of checklist of things or skills that we need to acquire during our clerkship so that we can seek out these opportunities even more…"*

Among the external resources that were indicated by the learners was the longitudinal theme of the resilience skills' building course.

*10M: "…it was good to take a break from all the clinical and scientific things that we were studying. It was a place you could go, and switch off and relax, you know, just unwind…"*

## Added value of the course

This theme includes the fragments of texts that show what the learners' perceive as the strengths of the course.

*4F: "…I think the setting helped a lot…sitting on bean bags; I was part of the second group, my batchmates from the first group kept raving about the fact that we get to sit on bean bags and relax instead of just remaining on chairs in our lectures all day…"*

**Space.** The first category within this theme sheds light on how in contrary to their other learning experiences, the learners considered the innovative nature of this course to offer them a breather.

*6F: "…I found the guided meditation and the visualization sessions before the exam really nice and useful…"*

According to the learners, fun was often experienced. They describe the course as a 'breather'. The learners valued how the setting was different than what they were used to and allowed for the release of tension.

*1M: "…it was fun… our Thursdays were packed with radiology and theoretical sessions of the rotation we are doing, so when we had the resilience skills' building course session, it was kind of like a break, which was also fun and informative..."*

The learners appreciated that the course was interactive, engaged them through a workshop style.

*1M: "…I enjoyed the sessions where we had 'to do something' like the ones where we needed to engage in the conversation or maybe when we had to mindfully eat the date or prepare our own gratitude jar… I think the session where we meditated was really nice. It was honestly the first time I tried this kind of thing. So, it was a new experience, it was nice, it was interesting, and it was something that I took back home…"*

They greatly valued that the course reassured them, and validated their internal and external experiences, normalizing matters.

*3M: "…it was kind of fun, different than the rest of the lectures we were supposed to attend…I benefitted from the reassurance integral to this learning experience… that you are okay, and it is okay to be this way…I feel I needed someone to tell me that it is okay…"*

**Substance.** The second category within this theme refers to the value that the learners reaped from the content offered.

> 3M: "…doing simple acts mindfully such as washing my hands or anything else that is really simple… I gained the capacity and willingness to really focus on those simple activities… for example, if I am washing my hands just before going into a surgery, or washing my hands before touching the patient to do a physical exam…the highlights were chewing the date mindfully, eating it really slowly. I think the breathing exercises, as well, where I learned how to relax my body…"

The learners described the experience as informative. They acquired knowledge and raised their level of awareness about the subject matter.

> 4F: "…It was so needed for us as medical students…for the future, raising our level of awareness about our own mental health (and that of others) has been so impactful; I am extremely grateful for having had this experience…"

The experience enabled the learners to acquire coping skills, including mindfulness and reflection.

> 1M: "…I learned a couple of skills which are useful… the meditation and mindfulness exercises..."

> 4F: "…the first skill that I gained is practicing mindfulness every day. For me, it was not a specific exercise that I needed to do. It was more like the skill became a part of how I am living… the second skill would be reflection…I could reflect not only on my intentions as to why I am doing what I am doing, but also how I can do it better…"

Some of the learners referred to the interconnection between the psychiatry rotation, which offered them technical knowledge, and the resilience skills' building course, which gave them insight and firsthand experience with complementary tools.

> 4F: "…it was really helpful to learn psychiatry while doing this course as well because you got to learn and apply coping mechanisms while acquiring relevant technical knowledge…"

## Sustainability of the effects of the course

This theme includes the fragments of texts that relate to the usability of what was acquired from the course.

**Shift.** This category includes the text fragments that show that the learners noticed internal shifts towards the better that they attributed to their experience of the resilience skills' building course.

The learners seemed to believe that the resilience skills' building course enabled them to develop healthy lifestyle habits, such as movement and mindfulness.

> 3M: "…the breathing exercises were helpful, and I do practice them. So, anytime that I feel that I am overwhelmed, I just take deep breaths, with the intention of relaxing myself, so that I can focus on the task-at-hand rather than feeling overly anxious…"

The resilience skills' building course also contributed to shifts in attitudes, according to the learners.

> *4F: "…exercising mindfulness, meditation, and positive affirmations; I have seen a change in my life due to these practices, and my batchmates, the ones I have spoken to, also agree with me so I would say it has been a really positive experience…"*

> *11F: "…for a lot of my friends, the course was something very new and they expressed that it completely changed their lives in terms of learning to cope much better…"*

The learners also reflected upon certain inner resources that they managed to get access to and develop through their engagement with the content of the resilience skills' building course.

> *6F: "…I started meditating and I find that very helpful, as well as just taking time for myself. I learned to prioritize myself at some point and to create a balance so that I can study. Before I sleep, I take 5 minutes to just reflect, take a deep breath, sometimes I do yoga so all of these really help me to sleep and rest. When I wake-up, it is a new day and that really helps…"*

Lastly, this category also encapsulated the fragments of text that show the learners' realizations around the rewiring and repatterning of their thinking habits.

> *2F: "…mindfulness sessions. It was so helpful… applying it to our lives has been so impactful…I am, of course, nowhere close to living mindfully, all the time. Yet, I now have more awareness. There is an inner voice in my head constantly reminding me to be mindful, to bring awareness to what I am doing, to do matters mindfully…"*

**Commitment.** It was apparent to the learners that they need to be committed to what they had acquired and integrated form the resilience skills' building course for its effects to be maintained over time. They had realized that they need to be responsible about their health and wellbeing, and need to exhibit a sense of ownership in this regard. They highlighted the requirement for active maintenance and focused attention.

> *3M: "…I have tried some of the techniques, but I failed at committing… For example, I tried keeping a diary like two or three times, somehow over time I end-up dropping the 'ritual' and focusing on other things or even forgetting about it or where I put the diary… doing simple exercises mindfully…I am not sure if I am doing it well; I have a tendency to feel agitated, to get distracted…It is sometimes hard to commit to staying mindful throughout an activity irrespective of its level of difficulty… especially when things are moving at a fast pace…"*

The learners also pinpointed the need to repeatedly make a conscious choice of deploying the acquired skills. This, to them, is especially relevant in challenging times, 'when the going gets tough'.

> *2F: "…I also picked-up the habit of journaling- it came in handy around the COVID-19 times…"*

> *9F: "…the real challenge is maintenance. I should have maintained certain skills that I had gained from the course. I feel like it is now, during the pandemic, that we are truly getting tested…"*

**Impact.** The learners indicated several long-term effects of the knowledge, skills, attitudes, and habits acquired from the resilience skills' building course.

*7M: "…the fact that most of us are saying that we should have the course throughout the year means that it has actually impacted a majority of us…"*

These effects include enhancing concentration, increasing efficiency, and improving quality-of-life.

*4F: "…living more mindfully enables me to do things in the same amount of time but in a way more relaxed manner…I now have a lot more clarity about what I am doing and where I am going… regularly reflecting enabled me to look back at my strategies of how I am doing things and instead of thinking: 'okay I just need to get this done', I say to myself: 'okay how can I do it better?' and then because of that, I think I have become a lot more efficient than I used to be…"*

*5F: "…I particularly liked the exercise of the gratitude jar. We were asked to write things we are grateful for. I took this concept into one of my rotations, as sometimes you focus so much on the negative and you forget the positive things in your life. So, for example, for one specific rotation, I remember it was for internal medicine, I added to my jar, using the extra coloured-papers that I kept from the session, all that I appreciated and was grateful for from that rotation. I consciously directed my attention away from the negative aspects of my rotation…"*

## Opportunities for improving the course

This theme includes the fragments of texts around how they think the resilience skills' building course can be improved. It mainly indicated the aspects that the learners appreciated and in turn wanted more of.

*5F: "…I think it would have been really good if we took some of the sessions outside the university especially the meditation sessions- it would be a nice change of air…"*

*11 F: "…meditation sessions in the park would be a really good change…"*

**Experientiality.** Besides valuing the engaging nature of the course, the learners suggested for there to be more interactions and socialization.

*1M: "…I favoured the sessions which were more interactive so I would say maximize the interactions throughout the course…the more interaction I had in the session, the more I took from the session…"*

They wanted for the resilience skills' building course to involve more doing, where they are taught 'how' more than 'what' and they are enabled to perform what they are getting taught during the respective course sessions.

*2F: "…let them be built in the format of a workshop and not a lecture. This would be better. What was special about the mindfulness sessions was that they felt more like workshops. There were bean bags, they were interactive activities, while attaining the goal of the lesson. So maybe sessions like the one concerning time management can become more workshop like…"*

*3M: "…I want it to be more workshop-like with more interactions… I think one of my coping mechanisms is to spend more time with my friends. Hence, I would rather spend more time with my friends instead of attending a lecture that tells me to spend more time with*

*my friends… if I was engaging with the content, at the same time engaging with the people around me, I will succeed in achieving these two things…"*

They wanted the learning experience to be even more relaxed, laid-back.

*3M: "…I do not want to think of this course as just another lecture…"*

*6F: "…Basically, for the last couple of years some of us have got used to just sitting and studying without much movement, maybe introducing some cardio sessions for our well-being would be helpful…"*

Lastly, within this category, the learners referred to the value of having theme-based support groups, where sharing of experiences are maximized.

*6F: "…I think we can maybe have a few sessions about specific issues that we might run into during our clinical placement (e.g., how to deal with the situation when a doctor is angry at something or when we have too much on our plate) …"*

*11F: "…I think it would be nice to have an online platform where people can anonymously talk about their experience in general, like a setting where people can share stories for example of how patients made a change in their lives or like how they coped with challenging situations in the clinic. We would feel less alone when we realize it is a shared experience…"*

**Configuration.** In terms of opportunities for improving the course, the learners reflected upon making the course optional, since they feel it is not necessarily for everyone.

*6F: "…the point about making the course optional I think is important because although several students are benefiting from it, unfortunately, it is not for everyone… if you make it optional, you can cater to those who really need it…"*

*8M: "…may be make it optional because I honestly feel it is not for everyone…. it did not really solve anything for me, but it might work for other people…"*

They also suggested making it available all year long.

*8M: "…I think maybe prolonging the course instead of making it only for half the semester, make it all though the semester…"*

*11F: "…definitely having the course throughout the year rather than just half the semester would be beneficial to see the actual effect… having the meditation sessions all year round would be a much better option…"*

The learners also suggested making it start earlier in their educational trajectory, ideally from the beginning of the program, since they believe they will be more receptive and become better prepared.

*3M: "…if such techniques were brought-up earlier in our medical education journey, there would have been noticeable changes… the earlier these coping mechanisms are instilled in us, the more impactful they would be…"*

*4F: "…I really want the content of this course to be available to students of years 1, 2, and 3 because I feel like if I had received this learning experience in the previous years, it would*

*have been extremely beneficial… it is more about raising their level of awareness early on. I do not think it is about causing noticeable change among them… it is to normalize feeling stressed or anxious as part of the medical education journey and shedding light on potential coping mechanisms…"*

## Output of integration

Integrating the outcomes of the quantitative and qualitative analyses revealed a comprehensive understanding of the situation, illustrated in the study's side-by-side joint display (Table 4). The merging of findings (i.e., primary inferences) enabled the development of a thorough understanding of how the resilience skills' building course affected the students' resilience and adaptability. As such, the integration led to seven meta-inferences: 'Students' vulnerability in periods of transition', 'Students' adaption to change', 'Course effect on students'

**Table 4. Seven meta-inferences resulted from the iterative joint display analysis process: 'Students' vulnerability in periods of transition', 'Students' adaption to change', 'Course effect on students' resilience', 'Students' engagement with the course content', 'Effects of COVID-19 on students', 'Students' adaptation to effects of COVID-19', and 'Course effect on students' resilience during COVID-19'. The secondary colour Green emerged by mixing the primary colour Blue with the primary colour Yellow (which constitutes an analogy of the critical thinking that took place to generate the meta-inferences from the integration of two sets of primary inferences). The integration led to confirming, expanding, and/ or refining of the researchers' overall understanding of the subject matter. It also led to the introduction of new insights.**

| Quantitative→ | Meta-inferences | ←Qualitative |
|---|---|---|
| A significant proportion of the students, at baseline, were at risk for burnout and anxiety, and would benefit from developing their resilience | **1. Students' vulnerability in periods of transition** (Confirmation and Expansion) | Students seemed to be well aware of the transitions that are integral to their educational journey, and the changes and challenges associated with, and their initial resistance to those transitions |
| There appeared to be an increase of professional efficacy over time | **2. Students' adaptation to change** (Expansion) | • Students seemed to be well aware of their capacity to adaptation to change, and the internal and external resources that they consciously/ subconsciously deploy in adapting to change <br> • Among the external resources that the students pointed-out as conducive to adaptation was the resilience skills' building course |
| There appeared to be no statistical differences in measures of burnout, anxiety, and resilience related to course delivery | **3. *Course effect on students' resilience*** (Refinement) | • Students highlighted the added value of the course (where it offered psychological space and deployable content) and the sustainability of its effects (including internal shifts to the better, requirement for commitment, and long-term impact) <br> • Students implied that the course not only offered them novel skills, it also gave them access to or activates their internal resources |
| – | **4. Students' engagement with the course content** (Introduction) | • Students highlighted their commitment and the efforts they direct towards deploying the skills acquired from the course to maintain their resilience over time <br> • Students also identified opportunities to improve the course, especially around experiential learning and course configuration |
| The overall risk for anxiety among students increased following the COVID-19 lockdown | **5. Effects of COVID-19 on students** (Confirmation and Expansion) | Students acknowledged the changes and challenges associated with COVID-19, and their resistance |
| | **6. Students' adaptation to effects of COVID-19** (Expansion) | • In terms of their adaptation, the students referred to resources that supported them during COVID-19 <br> ◦ Internal: <br> ▪ Accepting the factors that are out of their control <br> ▪ Becoming intentional about bridging gaps <br> ◦ External: <br> ▪ Belonging to community-of-practice <br> ▪ Sense of togetherness |
| | **7. *Course effect on students' resilience during COVID-19*** (Expansion) | Students highlighted the importance of commitment through repeatedly making conscious choice of deploying skills acquired from the course especially when the going got tough: COVID-19 |

resilience', 'Students' engagement with the course content', 'Effects of COVID-19 on students', 'Students' adaption to effects of COVID-19', and 'Course effect on students' resilience during COVID-19'.

First, the outcome of the qualitative analysis confirmed that the students were vulnerable, in terms of their mental health (upon starting the course). It also expanded the researchers' understanding by revealing the students' awareness of the transitions that are integral to their trajectory, and the associated perceived challenges and resistance. Second, the outcome of the qualitative analysis showed the students' awareness of how they tend to adapt to change, expanding on the quantitative finding that students' professional efficacy (as measured by MBI) increased over time. Third, although the quantitative outcome showed no significant effect of the course on students' resilience (which could be due to the limitations of the small sample size and/ or course duration), the qualitative outcome refined the researchers' understanding by revealing the students' point-of-view in relation to the added value of the course and the sustainability of its effects, and how it enabled the students to activate existing internal resources. Fourth, the outcome of the qualitative analysis introduced to the researchers how the students engaged with the course content, where the students highlighted their commitment towards exercising what they acquired and provided constructive feedback on means of improving the course. Fifth, the outcome of the qualitative analysis confirmed the effects of COVID-19 on the students and expanded the researchers' understanding by shedding light on the students' descriptions of their relevant lived experiences. Sixth, the outcome of the qualitative analysis also expanded the researchers' understanding of how the students' perceived themselves to have adapted to COVID-19 and its challenges. Seventh, the outcome of the qualitative analysis also expanded the researchers' understanding of how the students leveraged what they acquired form the course in their adaptation to COVID-19.

## Discussion

This study evaluated, in quantitative and qualitative terms, the implementation over time of a resilience skills' building course for undergraduate medical trainees. At baseline, upon the initiation of the students' clinical training, there appeared to be a significant number of students with indication of burnout and anxiety, similar to findings from other medical schools [105,106]. It was observed that, as the resilience skills' building course unfolded, the quantitative indices did not significantly change (which could be due to the limitations of the small sample size and/ or course duration). Yet, there appeared to be an overall increase in measurable anxiety after the unforeseen onset of COVID-19. The qualitative analysis revealed further insights into the positive effects of the respective course on the students and their adaptability. Di Vincenzo et al. (2024), in a recent systematic literature review, reported on levels of burnout among medical students measured using the MBI [64]. These levels varied widely from country to country, ranging from 5.6 to 88% (in the current study, 42.4% of participating students were at high risk for burnout, according to MBI). In the study of Di Vincenzo et al. (2024), the average burnout rate among students from 16 studies conducted in US was 44.2%, whereas the average burnout rate among students from 39 studies conducted in non-US countries was 23.9%. Predictors of burnout in these students included negative life events or poor motivation [64]. Another systematic literature review of 89 studies, by Ahmed et al. (2023), reported that approximately one third of undergraduate university students experience high levels of non-specific anxiety [65]. In that review, 38 studies focused on only undergraduate medical students, and only 4 studies were reported in the context of the COVID-19 pandemic. A variety of self-report questionnaires were used in these studies, with the overall prevalence in studies using the GAD-7 questionnaire reporting a pooled prevalence of 37.2% (95% CI

28.77-45.64) (in the current study, 34% of participating students were at high risk for anxiety, according to GAD). There was no consistent pattern of scores associated with the year of the study, the type of undergraduate education (medical versus other), or of socioeconomic variables, except that anxiety tended to be more common among females than among males. Multiple interventions have been tried for increasing resilience skills in health professions' students. Kunzler et al. (2020) reviewed 30 randomized controlled trials, of which 22 included only healthcare students. They found very low certainty evidence that compared with controls, these interventions were associated with reported higher levels of resilience, and lower levels of anxiety and stress perception, but no differences in depression, well-being, or quality of life [66].

The current study introduced a novel conceptual model: 'Resilience Skills' Building around Undergraduate Medical Education Transitions', which brings forth undergraduate medical students' perception about their adaptation to change and means of building their own resilience. This model can be deployed by medical educators to foster learning environments that inspire and empower students during transitions, whether those are integral or extrinsic to their trajectories. Operationalizing the model holds the potential of enabling students to thrive during challenges, maximizing their learning as they progress in their careers [70]. As suggested by the participating medical students, such a resilience skills' building theme can start earlier in their training and be optional (not mandatory), and can be designed in a way to become even more interactive.

The semi-quantitative tally around the output of the qualitative analysis revealed that the female student participants were more aware than the male student participants of the transitions that they had been going through, and the associated challenges and their resistance towards change. The female student participants were also more mindful of how they were adapting to the changes. The tally also showed how the female student participants were more cognizant of the added value of the respective resilience skills' building course, and the sustainability of its effects and opportunities for improving it. This may be related to the student participants emotional maturity which is associated with adult learning skill [107], and impacts students' academic performance, making it an important component of their professional development [108] and career adaptability [109]. This is in alignment with the literature around the subject matter which signifies that female medical students tend to be more sensitive and mature emotionally than male medical students [110].

This study showed that the students were vulnerable (and perceived themselves as such) in periods of transition. They seemed to become more prone to burnout and anxiety, and to benefit from proactively building their own resilience. It is established that medical students tend to be well aware of the persistent unique challenges that they face as part of their academic journey [111]. These challenges include a demanding curriculum, strenuous clinical rotations and placements, and patient care responsibility [111–113], contributing to heightened emotional distress [114]. This makes medical students vulnerable, and more prone to burnout and anxiety relative to enrolees in other programs [115,116]. In addition to the expected struggles of a medical students, the pandemic presented itself with unprecedented challenges to medical education; the shift to online education, disruption of clinical rotations, uncertainty around placements, and forced isolation all added to the preexisting stressors [117,118]. A previously conducted study showed that medical students tend to believe that if this vulnerability is not addressed, it could compromise their quality-of-life, affecting them personally as well as professionally [119]. There is an understanding that emotional vulnerabilities can be a risk factor on their academic progress [120]. Moreover, there tends to be cognizance among medical students that building resilience is likely to improve their quality-of-life, where they believed that it reduces symptoms of anxiety and stress and improves their overall health [121,122].

The study showed that the students were aware of their intrinsic capacity to adapt to change, and the benefits of deploying internal and external resources in that direction. This finding is in alignment with that generated from previously conducted investigations around self-reported adaptability of postgraduate dental learners due to abrupt transition to distance learning induced by the pandemic [123]. In the respective study, the learners perceived themselves to have adapted well to the transition; self-regulated learning appeared as a cornerstone in their adaptation to the accelerated change accompanying COVID-19. The results also revealed an interplay between the cognitions, emotions, and behaviours on the level of the self as part of the adaptation process. Another study capturing medical students' perception of an innovative co-curricular program showed how the students consider intrinsic attributes such as self-awareness, along with system and critical thinking, foresight into potential upcoming steps, and a growth mindset to enable them as they progress in their careers [124]. In that study, the students also frequently alluded to how going through enriching experiences (e.g., learning through traveling), along with receiving other forms of external support, raise their resilience (including but not limited to self-awareness) and in turn facilitate transitions integral to their educational trajectory.

Among the external resources that the students, who participated in the current study, identified as enablers, in times of transition, was the resilience skills' building course. They considered the respective course to offer them psychological space and deployable content, and contributed to internal shifts (to the better). There was a collective belief that the course will have positive long-term impact, and that sustaining its value will require commitment from their end. Medical students tend to value such curricular attempts aimed at building their resilience, where they consider it as a springboard to learn resilience skills. It was previously suggested that medical students appreciate the practical approach to skill building as well as the interactive shared experiences to such courses, especially during the clerkship years which are considered as a relatively isolating period during medical education [125]. The students participating in the current study implied that the course not only offered them novel skills, it also enabled them to activate existing internal resources. Relevantly, it is assumed that adult learners have mental models developed from previous experiences that form an increasing resource for learning; they learn through adapting those mental models [42,44]. This shows how building resilience and learning experientially can happen hand-in-hand.

Developing the course through design-based research enabled properly understanding the problem at hand (i.e., vulnerability of undergraduate medical students at times of transitions) and proactively addressing it (i.e., developing, implementing, and evaluating a curricular learning and teaching intervention that equips students with the skills to proactively build their own resilience) [58]. The iterative process, integral to design-based research, allowed for the optimization of the educational intervention design. In the context of the current study, design-based research also enabled the attainment of both practical and theoretical outcomes [57], similar to previously reported upon learning and teaching interventions that were developed using the same iterative technique [126]. Besides its practical value on the students, the firsthand experiences of the course reported upon in the current study revealed the value that resulted from anchoring the intervention not solely on Kolb's experiential learning theory (which does not capture the learning that occurs in relating to others) [45,46] but rather finding the complementarity of this valuable theory with a social constructionism theory (where a small group of people learn through their social interactions) [56]. Anchoring a learning and teaching intervention on the complementarity between Kolb's experiential learning theory and a social constructionism theory had been previously reported upon [69,124,126,127]. Through this complementarity experiential learning is pulled back to its origins, where it stemmed from human relations' training [48], and conceptualizes experiential education in

more sociological terms, showing that the individual learner is inevitably connected to social, cultural, and/ or environmental factors [52]. It also emphasizes that participation and learning go together, and the learner is embedded in the context of learning.

Using design-based research to co-create the course also allowed for seeking theoretical understanding throughout the research and development process, where the current study constitutes the second investigation in a series of research studies aimed at leveraging the practical, firsthand experiences to contribute to the knowledge base of the subject matter. The preceding study [70], as previously mentioned in this manuscript, was conducted to explore the perception of the students regarding their understanding of, and personal experience with building resilience, and their engagement with the content of the course. The findings of the current study (along with those generated from the preceding investigation) can enable transferring the resilience skills' building course to contexts similar to that of MBRU. The students' insights about the added value of the course (e.g., that it is reassuring and informative), and the sustainability of its effects (including but not limited to the requirement of active maintenance of acquired skills) and opportunities for improving it (around making it even more experiential and elongating the offering), will support in the transference of the offering to other settings and contextualizing it. These insights have broader implications for integrating resilience training into medical curricula.

It was evident from the current study that the students effectively engaged with the course content. They kept on referring to the efforts that they tend to direct towards deploying the skills acquired from the course to maintain their resilience over time. The participating students were aware that, especially when the going got (really) tough during the pandemic, they needed to repeatedly make the conscious choice of deploying the skills acquired from the course. Relevantly, the challenges to proactively building resilience that the students, in the preceding qualitative investigation around the same resilience skills' building course identified, include: personal resistance to changing old habits and adapting new ones, doubts around the efficacy of the exercises, prioritizing practicing the exercises, lack of internal motivation towards making active changes, and requirement of hard work [70]. According to the current study, the students proactiveness about identifying and communicating opportunities to improve the course is also indicative of their engagement. Similarly in the preceding study, one of three identified themes from the inductive qualitative analysis of students' reflections was 'Appraisal'; it includes highlighting aspects of the course that they appreciated (the assignments, development of a common ground/ a sense of togetherness, and acquired skills and noticeable changes in attitudes). The students actually expressed valuing actively engaging in the course and with its content [70].

In fact, the occurrence of COVID-19 amidst the study constituted, from an investigatory perspective, an opportunity to understand how a force majeure could affect undergraduate medical students as they are getting accustomed to the clinical environment. The effects of this major international crisis on the students could be perceived as an intensified version of what they go through as part of the changes expected in their trajectories. The perception of the same group of students, among the rest of the enrolees in the same MBBS program (along with their instructors), was captured in another investigation that evaluated the rapid transition to distance learning due to the onset of the pandemic [128,129]. The participating students, across all cohorts, expressed satisfaction with the transitioning (with a percentage of total average of 73%). The perceived readiness to transition to the clinical years was low (on a zero to ten scale) with a mean of 3.97(and a standard deviation of 2.32) [129].

It is worth noting that in parallel to the resilience skills' building course, the students participating in the current study were offered various web-based extracurricular activities (organized by the entity overseeing the students' affairs at MBRU, together with the students' council) to support and maintain a sense of community among the student body (which was

not limited to the student participants of the current study). Moreover, the participants of the current study, along with the rest of the student body and faculty, had access to a series of relaxation sessions. A peer-mentoring program was also implemented. Given that students in Phase 3 could not return to their clinical placements, part of their schedule was freed up, and volunteers were recruited to tutor students, especially those in Phase 1. This supported 'junior' students and provided 'senior' ones (among whom were enrolees of the course under investigation in the current study) with a sense of purpose [128]. All these concurrent supporting initiatives, during the pandemic, might have acted as confounders in the current study. According to the students participating in the current study, their adaptation to the effects of COVID-19 also, similar to other changes, required the deployment of internal and external resources. The internal resources that the students seem to notice the most (within the context of COVID-19) were first, exercising acceptance around factors that are out of their control, and second, becoming intentional about bridging gaps generated due to the pandemic. In terms of the external resources, the ones that seem most prominent to the students were first, belonging to a community-of-practice, and second, the sense of togetherness.

This study has a few limitations. Although selecting mixed methods research design allowed for the generation of transferable, in-depth insights, the generalizability of the findings is limited to universities that are contextually similar to MBRU. Resorting to focus group sessions, instead of interviews, allowed for rich discussions and valuable interactions among the participants. Yet, choosing focus group as the qualitative data collection tool limited the number of participants, and might have introduced biases (e.g., social desirability, groupthink, dominance, and/ or moderator bias). The sample size was small not only in the qualitative component of this study but also in the quantitative component covering a single cohort of undergraduate medical students only. As such, further testing of greater numbers of students over time in association with the resilience skills' building course is needed, diversifying the data collection tools given the potentiality of the abovementioned biases (along with the possible self-report bias associated with how burnout, anxiety, and resilience were quantitatively measured as part of the current study). It will be worthwhile for future studies to include multiple medical colleges, and/ or several cohorts within the same college.

## Conclusion

This study indicates that a resilience skills' building course as part of undergraduate medical education may not instantly affect student ratings of burnout, anxiety, and resilience. However, students likely engage with such a course and its content to acquire and deploy skills to cope with circumstances and to adapt to changes integral to their training or otherwise. This is especially true when the course is developed, implemented, and evaluated in alignment with the principles of design-based research, and anchored in constructivism experiential learning theories and designed to foster self-directed learning. As such, the findings encourage further developing and implementing contextualized learning opportunities for building resilience skills among future healthcare workers. In addition to fostering resilience skills, medical schools should also consider additional measures, such as directly addressing factors that cause internal distress among students, in order to improve their overall well-being.

## Supporting information

**S1 File. Appendix I Focus Group Protocol.**
(DOCX)

**S2 File. Raw Data.**
(DOCX)

## Author contributions

**Conceptualization:** Samuel B. Ho, Reem AlGurg, Adrian Stanley, Laila Alsuwaidi.

**Data curation:** Farah Otaki, Samuel B. Ho.

**Formal analysis:** Farah Otaki, Samuel B. Ho, Bhavana Nair, Amar Hassan Khamis.

**Investigation:** Samuel B. Ho.

**Methodology:** Farah Otaki, Samuel B. Ho, Amar Hassan Khamis, Agnes Paulus.

**Project administration:** Samuel B. Ho, Bhavana Nair, Reem AlGurg.

**Resources:** Samuel B. Ho.

**Supervision:** Samuel B. Ho, Adrian Stanley, Laila Alsuwaidi.

**Validation:** Farah Otaki, Samuel B. Ho.

**Visualization:** Farah Otaki, Samuel B. Ho.

**Writing – original draft:** Farah Otaki, Samuel B. Ho, Bhavana Nair.

**Writing – review & editing:** Farah Otaki, Samuel B. Ho, Reem AlGurg, Adrian Stanley, Amar Hassan Khamis, Agnes Paulus, Laila Alsuwaidi.

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
