## [Decision Letter · Decision Letter 0]

22 Dec 2024

PONE-D-24-31493Effects of building resilience skills among undergraduate medical students in a multi-cultural, multi-ethnic setting in the United Arab Emirates: a convergent mixed methods studyPLOS ONE

Dear Dr. Ho,

Thank you for submitting your manuscript to PLOS ONE. After careful consideration, we feel that it has merit but does not fully meet PLOS ONE’s publication criteria as it currently stands. Therefore, we invite you to submit a revised version of the manuscript that addresses the points raised during the review process.

We look forward to receiving your revised manuscript.

Kind regards,

Ashraf Atta Mohamed Safein Salem

Academic Editor

PLOS ONE

Journal Requirements:

2. In the ethics statement in the Methods, you have specified that verbal consent was obtained. Please provide additional details regarding how this consent was documented and witnessed, and state whether this was approved by the IRB.

3. In the online submission form, you indicated that relevant data are within the manuscript and its Supporting Information files. Complete data is available upon reasonable request.

Additional Editor Comments:

Below is a summary of the most common and important points that the authors should address to enhance the quality and clarity of the study.

1. Introduction

• Burnout and Related Psychopathologies:

o Include a brief definition of Burnout Syndrome (Line 45).

o Provide a stronger rationale linking burnout to other mentioned mental health issues, such as depression, stress, and anxiety, in the context of medical students.

o Avoid using the term "soft skills" for communication and empathy (Lines 66–67), as these are teachable, measurable, and significant skills.

• Theoretical Frameworks:

o Expand on the theoretical foundation, including the role of Social Constructionism Theory and Kolb’s theory, and discuss their relevance to resilience and medical education (Lines 81 and 92).

o Provide additional examples of how these theories have been applied in similar contexts.

• Global Context:

o Strengthen the statement about improving medical students' well-being by citing studies from diverse cultural and geographic contexts (Line 68).

• Qualitative Research in the Study:

o Briefly explain the use of qualitative methods in the Introduction to set the stage for the methodology.

2. Methodology

• Sampling:

o Provide more details on the sampling method, including:

 Criteria for participant selection.

 Verbal consent process and whether participation was optional (Line 231).

• Instrumentation:

o Address the validation of the instruments used (e.g., Connor-Davidson Resilience Scale, Maslach Burnout Inventory) for students in the MENA region.

o Justify the appropriateness of tools like the GAD-7 for this study and describe their relevance to cultural and demographic contexts.

• Data Collection:

o Add dates to Figure 2 for clarity about data collection timelines.

o Address any confounding factors such as differences in clinical exposure and support during the pandemic between groups.

• Qualitative Data Analysis:

o Avoid describing qualitative data collection as "exploratory" to prevent confusion about the convergent mixed-methods design (Line 270).

o Clarify NVivo's role in data analysis versus manual analysis.

3. Results

• Presentation:

o Reduce the number of text fragments/examples presented for each theme and move additional excerpts to an annex.

o Consider rephrasing themes like "suboptimal attitudes" to avoid judgmental language (e.g., "barriers to change").

o Clarify the meaning of key terms (e.g., "Adaption" vs. "Adaptation").

• Statistical Results:

o Discuss potential reasons for non-significant results (e.g., sample size, course duration).

o Provide more details on the statistical significance of Figure 5.

• Themes and Interpretation:

o Expand on critical themes such as interactivity, commitment, and experientiality, highlighting how they shaped students’ learning experiences.

o Discuss differences in anxiety and resilience levels between pre-clinical and clinical students.

4. Discussion

• Relevance of Theories:

o Strengthen the argument for applying Social Constructionism and Kolb’s theories, connecting them more directly to the study's findings.

• COVID-19 Context:

o Emphasize that the pandemic was a significant crisis, not merely a transition, and its impact on anxiety and resilience measures.

o Consider discussing confounding factors such as lockdown support or students’ pre-existing anxiety about clinical settings.

• Implications:

o Offer actionable recommendations for practice, such as making the course optional or integrating more interactive components.

o Discuss broader policy implications for integrating resilience training into medical curricula.

• Emotional Intelligence:

o Avoid discussing Emotional Intelligence (EI) unless directly supported by the study’s data, as this opens a contentious area of debate (Lines 605–607).

5. Limitations

• Methodological Constraints:

o Acknowledge limitations such as:

 Small sample size.

 Biases in self-reported measures.

 Challenges of using pre-developed scales with diverse cultural groups.

• PLS-SEM Limitations:

o Discuss the lack of confirmatory power in the model and its reliance on explanatory measures like R-squared values.

6. Figures and Formatting

• Improve the readability of Figures 3 and 5.

• Ensure consistency in terminology and grammatical clarity throughout the manuscript.

• Reduce the overall word count by summarizing repetitive sections and consolidating information.

Conclusion

Addressing these key points will significantly strengthen the manuscript by improving its clarity, rigor, and relevance. The suggestions for refining the theoretical framework, methodology, and discussion will enhance its impact and ensure it provides actionable insights for medical education. The authors are encouraged to carefully respond to the reviewers' comments and incorporate these changes systematically.

Reviewers' comments:

Reviewer's Responses to Questions

**Comments to the Author**

1. Is the manuscript technically sound, and do the data support the conclusions?

Reviewer #1: Yes

Reviewer #2: Yes

Reviewer #3: Yes

2. Has the statistical analysis been performed appropriately and rigorously? 

Reviewer #1: Yes

Reviewer #2: Yes

Reviewer #3: Yes

3. Have the authors made all data underlying the findings in their manuscript fully available?

Reviewer #1: Yes

Reviewer #2: Yes

Reviewer #3: Yes

4. Is the manuscript presented in an intelligible fashion and written in standard English?

Reviewer #1: Yes

Reviewer #2: Yes

Reviewer #3: Yes

5. Review Comments to the Author

Reviewer #1: REVIEW:

Full Title: Effects of building resilience skills among undergraduate medical students in a multicultural, multi-ethnic setting in the United Arab Emirates: a convergent mixed methods study

Manuscript Number: PONE-D-24-31493

Decision: Minor revision.

I congratulate the authors on the excellent work aimed at investigating the value of an innovative curriculum in developing resilience skills among undergraduate medical students in the United Arab Emirates. The study can contribute to the development of programs focused on skill training for healthcare professionals.

The manuscript presents a robust qualitative analysis that provides a clear overview of the students' perceptions regarding the proposed approach and the challenges faced during their academic journey.

In light of the above, I suggest a minor revision of the manuscript regarding the factors listed below:

Line 45 - I suggest including a brief definition of Burnout Syndrome.

The first paragraph of the Introduction mentions Burnout. The second paragraph refers to other mental disorders such as depression, stress, and anxiety. I suggest presenting a rationale that links all the mentioned psychopathologies.

Line 68 – “Universal efforts are needed to improve medical students’ well-being.”

I suggest presenting some studies related to different cultures/countries to support the statement in question.

Lines 79-80 – “This fortifies new knowledge, and changes patterns of behaviours leading to long-term change in practices.”

It would be important to mention the reference that supports this conclusion.

Line 81 - How might the limitation presented in Kolb's theory affect the study's objective, considering that the practice of Medicine involves human relationships?

Line 92 - I suggest providing a more comprehensive theoretical foundation for the “Social Constructionism Theory”, mentioning how both approaches can be applied to medical students.

Introduction - The Introduction of the manuscript could benefit from including a brief explanation of the qualitative research applied to the study.

Lines 191-193 – “The subjects covered as part of this course include: introduction to cognitive behavioural therapy, mental toughness, practicing mindfulness, emotional intelligence, coping strategies to increase personal resilience, and time management.”

I suggest providing a brief description of the topics that were part of the course curriculum. The reader may not be familiar with some of the approaches mentioned.

Results:

Lines 441-443 – “Overall, the number of students with high risk for anxiety (GAD score >10) increased from 34% at baseline to 53% at the last timepoint (test 4), which was during the COVID-19 related restrictions.”

I believe it is important to consider other factors, aside from the COVID-19 pandemic, that may have influenced the reported outcome.

Transitions

Lines 50-60 - I suggest exploring the narratives of participants 11F, 1M, and 2F regarding the category "Communication," as it is a recurring theme in all three excerpts.

Line 59 – Participant 2F - It is not clear in the presented excerpt that there was discomfort regarding the feedback directly from the physicians.

Line 73 – Participant 1M - It is not clear in the presented excerpt that the interaction with patients represented a challenge for the participant.

Line 91 – Participant 3M - I suggest specifying which difficulties, in the participant's view, were related to COVID-19.

Line 143 – There is no indication of self-confidence and continuous learning in the text fragment from participants 1M and 4F.

Line 163 – The text fragment from participant 5F does not present the challenges mentioned in line 162.

Lines 184-191 – Participant 4F/11F - The two text segments can also be classified as coping strategies in the learning journey.

Line 231 – Participant 3M - The excerpt only mentions the management of expectations, differentiating it from the concept of resilience.

Line 237 – Participant 1M - I suggest interpreting the segment based on the maximization of learning reported by the participant.

Line 304 – Participant 10M - The text segment does not reflect "resilience skills," as mentioned in line 303.

Line 317 – Participant 6F - The presented excerpt does not address that meditation and visualization techniques provide a break in the learning experience.

Lines 387-393 – Participant 4F/11F - The two text fragments can be interpreted as "learning strategies" focused on well-being.

Line 397 – Participante 6F - The presented text fragment can be interpreted as the "adoption of new habits," aiming for balance and well-being.

Lines 414 (Participant 3M) and Line 426 (Participant 9F) - The two fragments can be categorized as difficulties in maintaining skills.

Line 442 – Participant 5F - The text fragment does not report an increase in concentration. Gratitude is the focus of the mentioned excerpt.

Line 436 (Participant 4F) and Line 442 (Participant 5F) - The two segments did not portray an improvement in quality of life.

Line 557 – Discussion

I suggest relating and discussing the results of the studies presented in the first paragraph with the findings obtained in the current study.

Line 605 - How does the development of emotional intelligence in healthcare professionals relate to the current study?

Lines 638-640 - I suggest explaining how this excerpt relates to self-awareness, system and critical thinking, foresight into potential upcoming steps, and a growth mindset, mentioned in lines 636 and 637.

Lines 683-684 - I suggest presenting potential limitations when recommending the implementation of the program in other contexts.

Based on the observations of the participants' text fragments, I suggest reviewing the potential impact on the qualitative analysis presented as well as on the Discussion of the manuscript.

Reviewer #2: The authors could further elaborate on the implications of the non-significant statistical results. For example, discussing potential factors (such as sample size or course duration) that might have affected the outcomes would be beneficial. The qualitative analysis could provide good insights into the students' experiences. The researchers may consider expanding on how future curriculum improvements could benefit from these qualitative themes

The limitations need further discussion on the methodological constraints, for example (small sample size, potential biases in self-reported measures). Some sentences could be clearer with minor revisions to ensure grammatical accuracy and consistency in terminology.

Reviewer #3: Thank you for asking me to review this study. It is a very important topic, and a real effort given the circumstances at the time. The overall RCT design is an important element.

The introduction gives a clear rationale for the study, with suitable references relating to burnout.

I would avoid the use of the term ‘soft skills’ in relation to communication and empathy, since there is enough evidence that these are tangible, teachable and measurable. The term ‘soft skills’ tends to devalue those skills. Empathy in medical students demonstrably declines if not taught and reinforced throughout medical undergraduate programs. Having said that, empathy and communication skills appear as a non-sequitur in lines 66/67. I assume the authors are suggesting that developing empathy and communication skills reduces burnout and/or improves patient safety? If so, this needs clarification.

The design-based learning approach was a good idea for this intervention. The convergent mixed-methods design was suitable. However, I am not sure if the resilience course is experiential or contextualized learning based on the description. It is described to be a series of 6 hours of instruction. In the text fragments there is reference to sitting on beanbags/learning mindfulness. More detail on the course content and activities would be interesting to have in an annex.

The second group doing the course were working entirely online I suspect, during which there was no clinical exposure, while the first group were still in clinical placements. This could have either reduced or increased anxiety, depending on how nervous the students were about exposure to clinical settings, patients and uncertainty in the context of practising medicine. The amount of support students had during lockdown would also affect anxiety and resilience, and this is well described in the discussion. The additional supporting activities put on by the school during the pandemic could be confounders.

The authors talk at length about Kolb and constructivism; this could be expressed more succinctly. The course development process is described in a very thorough and clear way.

The baseline testing took place in November 2019, which did not coincide with Covid 19 pandemic. The second test took place presumably at the start of 2020, when the pandemic was beginning to show its face, but lockdowns were not yet in place. The crossover with second ‘baseline’ testing appears to happen 2 further months down the road, when the pandemic and lockdowns had begun, around Feb/March 2020. A final fourth test happened 2 months later, in May 2020 which coincided in Dubai with a gradual reduction of some restrictions.

The graphical display of the timeline of instruction and testing in Figure 2 is helpful. Between the abstract and the data collection description, it is slightly confusing, and I had to re-read both several times. Figure 2 should have dates of the data collection.

Figure 3 is unreadable even upon downloading.

Figure 5 would benefit from more detail on statistical significance.

Line 231: it states that ‘after all the students enrolled in both courses gave verbal consent’ to participate. Was there an option not to participate?

The GAD-7 is a well-recognized instrument in clinical and research use.

The Maslach Burnout Inventory is described as validated in numerous populations, which is important to mention. Does this include Arab/North African populations? Is the MBI validated for students in the MENA region?

I am not familiar with the Connor-Davidson Resilience scale. The authors do not mention how well validated this instrument is, and in what contexts it is used.

The use of the term exploratory in relation to the qualitative data might be slightly confusing since the study is also described as convergent (or concurrent). Eg sentence 270: ‘The qualitative data collection was exploratory’. The authors probably don’t need to say this. Simply describe that students participating in the focus groups were asked to talk about their lived experiences. This section about the qualitative data would benefit from being re-written more succinctly to improve clarity.

The methodology for data analysis is well described and widely accepted (Braun & Clarke). It is not completely clear how NVivo software was involved, as the description of the analysis appears to be entirely manual.

The method for joint display analysis is sound.

Results:

To what extent is the baseline level of anxiety reflective of the fact that students had only just joined clinical clerkships for the first time? The MBRU program is described as mainly didactic from years 1-3. Is there any information about levels of stress/burnout in pre-clinical students in the program around the same time?

I would recommend using only one or two examples per theme; there are too many examples. Please put most of the comments in an annex.

I am not sure if the ‘suboptimal attitudes’ theme is quite suitable: it comes across as judgmental. It might be better described as ‘barriers to change’

Are you sure about the use of the word ‘Adaption’ – do you mean adaptation? The word adaption is specifically from biology (eg evolutionary adaptions such as a giraffe’s long neck; not the same as a human’s adaptations to change within their lifespan).

The ‘Shift’ and ‘Experientiality’ themes struck me as most important. Here, there is reference to comments about interactivity being important to students. Please clarify how much of this course was interactive and how much didactic.

The ‘Commitment theme’ was especially interesting. There are comments from some students that this type of course is not for everyone and might be better as an optional opportunity. What do the authors think about this?

The authors put forward a model for Resilience skills building around transitions. The Covid 19 pandemic was more than a transition, it was a major international crisis. It is not particularly surprising that levels of anxiety rose or resilience measures did not change much. I agree totally with line 600, that gradual changes in emotional maturity in students contribute to their development which may be hard to unpick from measures of anxiety/resilience.

I would be a bit careful about introducing anything new such as emotional intelligence in the discussion, since there is considerable discussion about whether EI is a valid concept in the first place. Furthermore, although there are supposed measures of EI, these are arguable. I don’t think that lines 605 to 607 follow from lines 602 to 604. I would take out the sentence from 605 to 607 as it opens a Pandora’s box.

The conclusion is reasonable, although I think there is a bit of confusion around whether this is experiential/contextualized learning. Please clarify what context the authors are referring to: the campus experience, the clinical experience, students’ home lives or all of the above?

I suggest trying to cut down the word count. Many of the sentences could be shortened.

Many thanks again for asking me to review the paper.

6. PLOS authors have the option to publish the peer review history of their article (what does this mean?). If published, this will include your full peer review and any attached files.

Reviewer #1: **Yes: **Débora Regina de Aguiar

Reviewer #2: **Yes: **Prof Dr Saad S Alatrany

Reviewer #3: No

---

## [Author Response · Author response to Decision Letter 0]

14 Jan 2025

Comment Response

Academic editor:

Below is a summary of the most common and important points that the authors should address to enhance the quality and clarity of the study. Thank you for taking the time to summarize all the valuable feedback that we have received from all three reviewers. The lines’ referencing in our responses below, in Bold, is according to the manuscript with track changes (and not the ‘clean copy’ with the feedback already integrated).

1. Introduction

• Burnout and Related Psychopathologies:

o Include a brief definition of Burnout Syndrome (Line 45). As suggested by Reviewer 1, we looked into several definitions of burnout syndrome and chose to reflect in the beginning of the Introduction section that of Professor Christina Maslach (Lines 42 through 45), given (a) the prominence of her contribution in this realm and also that (b) one of the three scales that we used for the quantitative component of this study is the Maslach Burnout Inventory (MBI):

‘Burnout is a syndrome associated with long-term fatigue, physical exhaustion, hopelessness, and helplessness in people who are exposed to intense emotional demands due to their job, and who constantly engage with other people, with negative attitudes towards work, life, and/ or other people (1).’

o Provide a stronger rationale linking burnout to other mentioned mental health issues, such as depression, stress, and anxiety, in the context of medical students. We also added (in response to one of the comments of Reviewer 1) a sentence to describe three components of burnout (as characterized by Professor Maslach): emotional exhaustion, depersonalization, and low perceived personal accomplishment, as a way to link burnout syndrome to other psychopathologies (Lines 45 through 47):

‘In simple terms, burnout can be defined as emotional exhaustion, depersonalization, and low personal accomplishment seen in individuals who have an intense relationship with people as part of their job (2-5).’

We also referred to the following Systematic Literature Review that directly links burnout with both depression and anxiety (Line 67- reference number 16):

https://www.frontiersin.org/journals/psychology/articles/10.3389/fpsyg.2019.00284/full

o Avoid using the term "soft skills" for communication and empathy (Lines 66–67), as these are teachable, measurable, and significant skills. Thank you for highlighting this point. As per one of the suggestions of Reviewer 3, we changed ‘soft skills’ to ‘non-technical skills’, and in-between brackets described that we are referring to the skills that are complementary to those of basic and clinical medical sciences (Lines 72 through 74):

‘…by developing non-technical skills (complementary to those of basic and clinical medical sciences) required in the practice of medicine such as empathy and communication (24, 25).’

• Theoretical Frameworks:

o Expand on the theoretical foundation, including the role of Social Constructionism Theory and Kolb’s theory, and discuss their relevance to resilience and medical education (Lines 81 and 92). As per one of the comments of Reviewer 1, we have clarified (through several additions) how the respective resilience skills’ building course is built upon the complementarity between Kolb’s experiential learning theory and Social Constructionism theory:

Lines 94 through 96: ‘This is particularly relevant to the practice of medicine given its reliance on others, within the same health profession and inter-professionally, and also with patients and their families (49, 50).’

Lines 105 and 106: ‘Social constructivism proposes that knowledge is constructed through social processes and interactions (54, 55).’

Lines 686 through 698: ‘Besides its practical value on the students, the firsthand experiences of the course reported upon in the current study revealed the value that resulted from anchoring the intervention not solely on Kolb’s experiential learning theory (which does not capture the learning that occurs in relating to others) (45, 46) but rather finding the complementarity of this valuable theory with a social constructionism theory (where a small group of people learn through their social interactions) (56). Anchoring a learning and teaching intervention on the complementarity between Kolb’s experiential learning theory and a social constructionism theory had been previously reported upon (69, 121, 123, 124). Through this complementarity experiential learning is pulled back to its origins, where it stemmed from human relations’ training (48), and conceptualizes experiential education in more sociological terms, showing that the individual learner is inevitably connected to social, cultural, and/ or environmental factors (52).’

o Provide additional examples of how these theories have been applied in similar contexts. In the Discussion section, in response to Reviewer 1, we added examples of studies that report on learning and teaching interventions that are designed in alignment with the identified theories (Lines 692 through 694):

‘Anchoring a learning and teaching intervention on the complementarity between Kolb’s experiential learning theory and a social constructionism theory had been previously reported upon (69, 121, 123, 124).’

• Global Context:

o Strengthen the statement about improving medical students' well-being by citing studies from diverse cultural and geographic contexts (Line 68). The corresponding feedback was quite valuable.

As per the feedback of Reviewer 1, we did a thorough review of relevant interventions implemented across the world, and referred to a selection of the ones that proved effective for improving measures of student wellbeing. These have been referenced in the Introduction section (Lines 75 through 78), and include studies revolving around mindfulness-based approaches, stress management and self-care training, communication skills training, and small group curricula.

• Qualitative Research in the Study:

o Briefly explain the use of qualitative methods in the Introduction to set the stage for the methodology.

 We value this feedback from Reviewer 1.

We have clarified in the Introduction section (Lines 136 through 140) the application of qualitative research as part of mixed methods, where we highlight the link between the three research questions and the corresponding components of the study (one of which is the qualitative analysis). As such, the current study is not purely qualitative (but rather mixed methods, with three equally important components- quantitative, qualitative, and integration), the Methods section describes those three components (discretely), and the choices around reporting on the study as a whole for consistency purposes follow the same guidelines of mixed methods research:

https://psycnet.apa.org/fulltext/2018-00750-003.html

2. Methodology

• Sampling:

o Provide more details on the sampling method, including:

 Criteria for participant selection.

 Verbal consent process and whether participation was optional (Line 231). This feedback, from Reviewer 3, pointed out an ambiguity in how we were reporting on the mechanism of the quantitative data collection; we have modified the corresponding section (Lines 234 through 237) to more accurately describe how the corresponding data was collected:

• Inclusion criterion was enrolment in 4th year

• Students’ participation was completely voluntary

• Students gave verbal informed consent to participate in the study

• Instrumentation:

o Address the validation of the instruments used (e.g., Connor-Davidson Resilience Scale, Maslach Burnout Inventory) for students in the MENA region. In response to this feedback from Reviewer 3, we identified several instances where the respective tools got validated in the MENA, in general, and UAE, in specific, and referred to them in the corresponding section of the current write-up (Lines 264 through 289).

o Justify the appropriateness of tools like the GAD-7 for this study and describe their relevance to cultural and demographic contexts. We highlighted (in response to Reviewer 3) that GAD-7 is a standard, validated tool for measuring anxiety, and has been tested in numerous cultural and demographic settings in the Middle East (Lines 272 through 274).

• Data Collection:

o Add dates to Figure 2 for clarity about data collection timelines. In response to Reviewer 3, we added an indication of the timespan in the caption of Figure 2 (Lines 260 through 262).

o Address any confounding factors such as differences in clinical exposure and support during the pandemic between groups. As per the suggestions of Reviewer 2 and Reviewer 3, we reflected on potential confounders in the Discussion section:

Lines 754 through 756: ‘All these concurrent supporting initiatives, during the pandemic, might have acted as confounders in the current study.’

• Qualitative Data Analysis:

o Avoid describing qualitative data collection as "exploratory" to prevent confusion about the convergent mixed-methods design (Line 270). In response to Reviewer 3, we removed the word ‘exploratory’ from the Data Collection subsection (Line 292).

o Clarify NVivo's role in data analysis versus manual analysis. In response to Reviewer 3, the corresponding section has been clarified (Lines 358 through 361).

The process was manual following the Braun and Clarke six-step framework. NVivo software was used as a facilitator only to assign codes to the sub-categories, categories, and themes (i.e., tagging), which enabled the classification of the text fragments (as per the conceptual model that was generated from the manual qualitative analysis).

3. Results

• Presentation:

o Reduce the number of text fragments/examples presented for each theme and move additional excerpts to an annex. We removed some of the selected exemplars and shortened a few others (as per the feedback of Reviewer 1), Lines 1 through 531 of the Output of qualitative analysis subsection of the Results section. The reporting on qualitative results is as per the corresponding guidelines of mixed methods research, which is referred to in the article:

https://psycnet.apa.org/fulltext/2018-00750-003.html. Those guidelines require for the reporting on qualitative results in mixed methods research to be narratively done. The selected text fragments are only a small proportion of the text fragments that underwent the six-step analysis. All the text fragments are provided in the supplementary document, with clear indication of which of the fragments were incorporated into the Results section.

o Consider rephrasing themes like "suboptimal attitudes" to avoid judgmental language (e.g., "barriers to change"). As per one of the comments of Reviewer 3, we slightly modified the description of this code in the context of the study to address the corresponding ambiguity (Lines 98 through 100 of the Output of qualitative analysis subsection of the Results section). We wish, though, to retain the critical language. Note that we are not judging the students. In phenomenological thematic analysis, we are supposed to empathize with the participants, to ‘put ourselves in their shoes’. With this code, we are trying to convey their evident self-awareness that initially their attitude was suboptimal. This is key to our storyline.

o Clarify the meaning of key terms (e.g., "Adaption" vs. "Adaptation"). As per one of the comments of Reviewer 3, we have clarified that we are using the term ‘adaptation’ rather than ‘adaption’. The word ‘adaptation’ is used because the expressions of the students show that they (proactively) deploy resources, they make conscious choices to cope (Lines 117 through 119 of the Output of qualitative analysis subsection of the Results section).

• Statistical Results:

o Discuss potential reasons for non-significant results (e.g., sample size, course duration). As Reviewer 2, we added those potential justifications (Lines 574 through 576):

‘It was observed that, as the resilience skills’ building course unfolded, the quantitative indices did not significantly change (which could be due to the limitations of the small sample size and/ or course duration).’

o Provide more details on the statistical significance of Figure 5. In response to Reviewer 3, we added the following sentence to the caption of Figure 5 (Lines 462 and 463):

‘…with no statistically significant difference across the 4 timepoints’.

• Themes and Interpretation:

o Expand on critical themes such as interactivity, commitment, and experientiality, highlighting how they shaped students’ learning experiences. In response to Reviewer 2 and Reviewer 3, we have expanded on these findings in the Discussion section (Lines 708 through 714):

‘The students’ insights about the added value of the course (e.g., that it is reassuring and informative), and the sustainability of its effects (including but not limited to the requirement of active maintenance of acquired skills) and opportunities for improving it (around making it even more experiential and elongating the offering), will support in the transference of the offering to other settings and contextualizing it. These insights have broader implications for integrating resilience training into medical curricula.’

o Discuss differences in anxiety and resilience levels between pre-clinical and clinical students. All of the students in the study were in their first clinical year. Pre-clinical students were not included.

4. Discussion

• Relevance of Theories:

o Strengthen the argument for applying Social Constructionism and Kolb’s theories, connecting them more directly to the study's findings. As per the feedback from Reviewer 1, we elaborated on the theoretical foundation of the intervention in both the Introduction and Discussion sections.

Lines 94 through 96: ‘This is particularly relevant to the practice of medicine given its reliance on others, within the same health profession and inter-professionally, and also with patients and their families (49, 50).’

Lines 105 and 106: ‘Social constructivism proposes that knowledge is constructed through social processes and interactions (54, 55).’

Lines 686 through 698: ‘Besides its practical value on the students, the firsthand experiences of the course reported upon in the current study revealed the value that resulted from anchoring the intervention not solely on Kolb’s experiential learning theory (which does not capture the learning that occurs in relating to others) (45, 46) but rather finding the complementarity of this valuable theory with a social constructionism theory (where a small group of people learn through their social interactions) (56). Anchoring a learning and teaching intervention on the complementarity between Kolb’s experiential learning theory and a social constructionism theory had been previously reported upon (69, 121, 123, 124). Through this complementarity experiential learning is pulled back to its origins, where it stemmed from human relations’ training (48), and conceptualizes experiential education in more sociological terms, showing that the individual learner is inevitably connected to social, cultural, and/ or environmental factors (52).’

• COVID-19 Context:

o Emphasize that the pandemic was a significant crisis, not merely a transition, and its impact on anxiety and resilience measures. As per Reviewer 3 suggestion, we modified the write-up to emphasize the significance of the pandemic (Lines 732 through 737).

‘In fact, the occurrence of COVID-19 amidst the study constituted, from an investigatory perspective, an opportunity to understand how a force majeure could affect undergraduate medical students as they are getting accustomed to the clinical environment. The effects of this major international crisis on the students could be perceived as an intensified version of what they go through as part of the changes expected in their trajectories.’

o Consider discussing confounding factors such as lockdown support or students’ pre-existing anxiety about clinical settings. As per the suggestions of Reviewer 2 and Reviewer 3, we mentioned potential confounders.

Lines 754 through 756: ‘All these concurrent supporting initiatives, during the pandemic, might have acted as confounders in the current study.’

• Impli

---

## [Editor Report · Decision Letter 1]

29 Jan 2025

Effects of building resilience skills among undergraduate medical students in a multi-cultural, multi-ethnic setting in the United Arab Emirates: a convergent mixed methods study

PONE-D-24-31493R1

Dear Dr. Samuel B. Ho,

We’re pleased to inform you that your manuscript has been judged scientifically suitable for publication and will be formally accepted for publication once it meets all outstanding technical requirements.

Kind regards,

Ashraf Atta Mohamed Safein Salem

Academic Editor

PLOS ONE

Additional Editor Comments (optional):

Thank you for your responses, the revisions made to the manuscript have significantly enhanced its clarity, rigor, and overall quality. The additional data/analysis and explanations you provided have strengthened the manuscript and ensured that it meets the journal’s standards.
---

## [Editor Report · Acceptance letter]

PONE-D-24-31493R1

PLOS ONE

Dear Dr. Ho,

I'm pleased to inform you that your manuscript has been deemed suitable for publication in PLOS ONE. Congratulations! Your manuscript is now being handed over to our production team.

Kind regards,

on behalf of

Dr. Ashraf Atta Mohamed Safein Salem

Academic Editor

PLOS ONE